physical chemistry

critical micellar concentration, DTAB-rich, SDS-rich, surface properties, thermodynamic properties

**Author for correspondence:**
Ajaya Bhattarai
e-mail: bkajaya@yahoo.com

This article has been edited by the Royal Society of Chemistry, including the commissioning, peer review process and editorial aspects up to the point of acceptance.

# Self-assembly of sodium dodecylsulfate and dodecyltrimethylammonium bromide mixed surfactants with dyes in aqueous mixtures

K. M. Sachin[1], Sameer A. Karpe[1], Man Singh[1] and Ajaya Bhattarai[1,2]

[1]School of Chemical Sciences, Central University of Gujarat, Gandhinagar, India
[2]Department of Chemistry, Tribhuvan University, M. M. A. M. Campus, Biratnagar, Nepal

(iD) AB, 0000-0002-2648-4686

The micellar property of mixed surfactant systems, cationic (dodecyltrimethylammonium bromide, DTAB) and anionic (sodium dodecylsulfate, SDS) surfactants with variable molar ratios in aqueous system has been reported by using surface tension and conductivity measurements at $T = 293.15$, 298.15 and 303.15 K. DTAB concentrations are varied from $1.0 \times 10^{-4}$ to $3 \times 10^{-4}$ mol l$^{-1}$ in $1.0 \times 10^{-2}$ mol l$^{-1}$ SDS solution while the SDS concentration is varied from $1.0 \times 10^{-3}$ to $1.5 \times 10^{-2}$ mol l$^{-1}$ in approximately $5.0 \times 10^{-3}$ mol l$^{-1}$ DTAB, so that such concentrations of DTAB-SDS (DTAB-rich) and SDS-DTAB (SDS-rich) solutions were chosen 3 : 1 ratio. The critical micellar concentration, as well as surface and thermodynamic properties for DTAB-rich and SDS-rich solutions, were evaluated by the surface tension ($\gamma$) and conductivity ($\kappa$) methods. The pseudo phase separation model was coupled with the dissociated Margules model for synergism. The Krafft temperature behaviour and optical analysis of mixed surfactants are studied using conductivity and UV–Vis spectroscopy, respectively. The dispersibility and stability of DTAB-rich and SDS-rich solutions with and without dyes ($2.5 \times 10^{-5}$ mol l$^{-1}$ of methyl orange and methylene blue) are carried out by using UV–Vis spectroscopy and dynamic light scattering.

# 1. Introduction

For the understanding of both fundamental and applicative prospects of mixed surfactant systems, the scrutiny of their various significant physico-chemical aspects becomes necessary. These physico-chemical aspects include aggregates formation, which depends on system environmental parameters including temperature and the other additives [1].

The physico-chemical properties (PCPs) of solutions containing both cationic and anionic surfactants, i.e. catanionic surfactant systems are different from those of their individual components. Due to the synergetic effect, the critical micellar concentration (CMC) value of mixed surfactant system is less than the pure surfactant system [2]. The catanionic surfactant system shows lower CMC, higher cloud point, lower Krafft point and higher surface activities; these properties are accommodating for their various applications [3]. At the appropriate concentrations and mole ratios, this catanionic surfactant system spontaneously tends to form fascinating structures, like mixed micelles [4], vesicles [5] and extended networks. The mixed surfactants play a very significant role in our life. It has several applications in the industrial and biological fields such as oil recovery enhancement, detergency, cosmetics, emulsification, solubilization, food industries, targeted drug delivery, wastewater treatment, chemical purification and synthesis of advanced nanomaterials [6,7]. Therefore, mixed surfactant systems are believed to be superior to single surfactant systems [8].

To date, many researchers have reported the study of micelle formation and aggregation process in the various solvent media [9,10]. The phase behaviour of various catanionic surfactant mixtures in aqueous solution is also well studied [11,12]. Among them, Zana *et al.* [13] studied the micellization behaviour of cationic surfactant mixtures with an aqueous solution; nevertheless, the thermodynamic properties of the solutions were not investigated. Sohrabi *et al.* [14] studied the phase behaviour and aggregate structures of mixed surfactants by using pulsed field gradient stimulated echo NMR technique. Yousefi *et al.* [15] reported the effect of co-solvent on the spontaneous formation of nanorod vesicles in catanionic mixtures of cetyltrimethylammonium bromide (CTAB) and sodium dodecylsulfate (SDS). Bhattarai [16] studied the micellization behaviour of cationic surfactant mixtures in aqueous solution as well as methyl alcohol–water mixture along with the detailed investigation of thermodynamic phenomena. Aslanzadeh & Yousefi [17] reported the micellization behaviour of tetradecyltrimethylammonium bromide (TDTAB) and SDS in the water–ethanol mixture. The self-assembling of catanionic surfactant mixtures in aqueous ionic liquid was studied by Sohrabi *et al.* [18]. The self-aggregation behaviour of catanionic surface active ionic liquids was studied by Xu *et al.* [19]. Similarly, the effect of the molecular structure of cationic surfactant mixtures on the interfacial properties at the oil–aqueous interface was reported by Wang *et al.* [20]. Earlier studies have been reported on the interaction of mixed surfactants with salts [21]. Recently, Bhattarai *et al.* [22] determined the micellar properties between DTAB and SDS in methanol–water mixed solvent media of varying proportions at 293.15 K. In this study, we have studied the effect of temperature on CMC and several other surface properties in depth; such type of study is not reported yet. Sachin *et al.* [23] have reported in the previous study on PCPs of DTAB-rich and SDS-rich mixed surfactant in the aqueous medium at 293.15, 298.15 and 303.15 K.

Our first objective is to determine the surface properties as the maximum surface excess concentration ($\Gamma_{max}$), area occupied per surfactant molecule ($A_{min}$), surface pressure at the CMC ($\pi_{cmc}$), the free energy of adsorption ($G_{ads}^o$), packing parameter ($P$), free energy of surface at equilibrium  ($G_{min}$) and the thermodynamic properties such as the degree of ionization ($\alpha$), the standard behaviour of Gibb's free energy of micellization ($\Delta G_m^o$), enthalpy ($\Delta H_m^o$), entropy ($\Delta S_m^o$), heat capacity of micellization ($\Delta_m C_P^o$) of SDS-rich and DTAB-rich from $\gamma$ and $\kappa$ data, respectively. The study of synergistic effects of our system with the help of models is also our interest. Moreover, we study the Krafft temperature behaviour and the optical analysis of mixed surfactants by conductivity measurements and UV–Vis spectroscopy, respectively.

Earlier studies have reported the effect of dyes with single and mixed surfactants [24,25]. Samiey & Ashoori [26] have studied the kinetic and thermodynamic properties on the effect of crystal violet with DTAB and SDS. Dey *et al.* [27] have also reported the diffusion rate of dyes with SDS and DTAB vesicle in bulk water separately. Therefore, dye-surfactant interactions studies are very useful for industrial applications, chemical research and dye separation processes [28]. Till now, there was no literature reported yet on the effects of methylene blue (MB) and methyl orange (MO) on equimolar concentrations of SDS-rich and DTAB-rich.

Our second objective is to see the effects of each concentration of DTAB-rich surfactant with MB and MO separately and also to determine the binding and distribution constants by using the spectroscopic method as well as to estimate the stability of SDS-rich and DTAB-rich surfactant with and without dyes from dynamic light scattering (DLS) analysis.

**Table 1.** Comparison of measured densities values ($\rho$) and surface tension ($\gamma$) values of DMSO at $T = 293.15$, 298.15 and 303.15 K with the literature data. Standard uncertainties are u($T$) = $\pm$ 0.01 K, u($p$) = $\pm$ 0.01 MPa. Unit: $\Delta\rho = 10^3$ kg m$^{-3}$, $\Delta\gamma$ = mN m$^{-1}$. $\Delta\rho$ = Exp. - Lit., $\Delta\gamma$ = Exp. - Lit. values.

| temperature (K) | $\rho/10^3$ kg m$^{-3}$ | | $\gamma$/mN m$^{-1}$ | | $\Delta\rho$ | $\Delta\gamma$ |
| | Exp. | Lit. [29] | Exp. | Lit. [30] | | |
|---|---|---|---|---|---|---|
| 293.15 | 1.100103 | 1.10073 | 43.41 | 43.36 | −0.000627 | 0.05 |
| 298.15 | 1.095079 | 1.09574 | 42.78 | 42.70 | −0.000661 | 0.08 |
| 303.15 | 1.090043 | 1.09074 | 42.12 | 42.05 | −0.000697 | 0.07 |

## 2. Material and methods

### 2.1. Materials

The DTAB (purity approx. 99%), SDS (approx. 98.5%), MO (greater than 85%) and MB (greater than 96%) used in the experiments were obtained from Sigma-Aldrich and SD Fine Chemicals Ltd (Mumbai, India). The surfactants were stores of DTAB and SDS, the vacuum desiccator filled with $P_2O_5$ due to their hygroscopic nature.

### 2.2. Methods

Milli-Q water was used for the preparation of all the solutions of mixed surfactants at three different temperatures as described in the earlier study [23]. Also, Milli-Q water was used for the preparation of the aqueous solutions of MB and MO and stored in the airtight volumetric flasks.

### 2.3. Density measurements

The densities ($\rho$) were measured by Anton Paar DSA 5000 M density meter, which was calibrated with DMSO (table 1) at $T = 293.15$, 298.15 and 303.15 K and compared with the literature [31].

### 2.4. Surface tension measurements

The electronic counter counted the pendant drop numbers of mixed surfactants with the Survismeter [31] after attaining a thermal equilibrium in Lauda Alpha RA 8 thermostat with $\pm$ 0.05 K. The reported surface tension was average values of three repeated measurements with $\pm$ 0.03 mN m$^{-1}$ combined uncertainty in surface tension. The Survismeter was calibrated by using DMSO. The surface tension data of our system agree well with the literature value (table 1) [30]. The presented surface tension ($\gamma$) and log $C$ (i.e. $C$ is the surfactant concentration) were plotted, for calculating the CMC value.

### 2.5. Conductance measurements

Specific conductance data were measured at $T = 293.15$, 298.15 and 303.15 K with the Pye-Unicam PW 9509 model conductivity meter having the frequency of 2000 Hz using a dip-type cell with a cell constant of 1.15 cm$^{-1}$ with an uncertainty of 0.01%. The instrument cell was calibrated by using the proposed method [32] using the aqueous potassium chloride solution. The temperature of the measurement cell was controlled with a Lauda Alpha RA 8 thermostat with $\pm$ 0.05 K.

### 2.6. UV–visible spectroscopy

Absorbance was measured by the Spectro 2060 plus model of UV–visible spectrometer. The spectral analysis was done in the range of 200–600 nm at 298.15 K. All UV–visible measurements were carried out with the following procedure. Firstly, the measurement of baseline with water was done. After that, 3 ml of DTAB-rich and SDS-rich different concentrations of surfactant solutions involved obtaining a well-marked absorption band. For mixed surfactants interaction with dyes, the baseline was registered

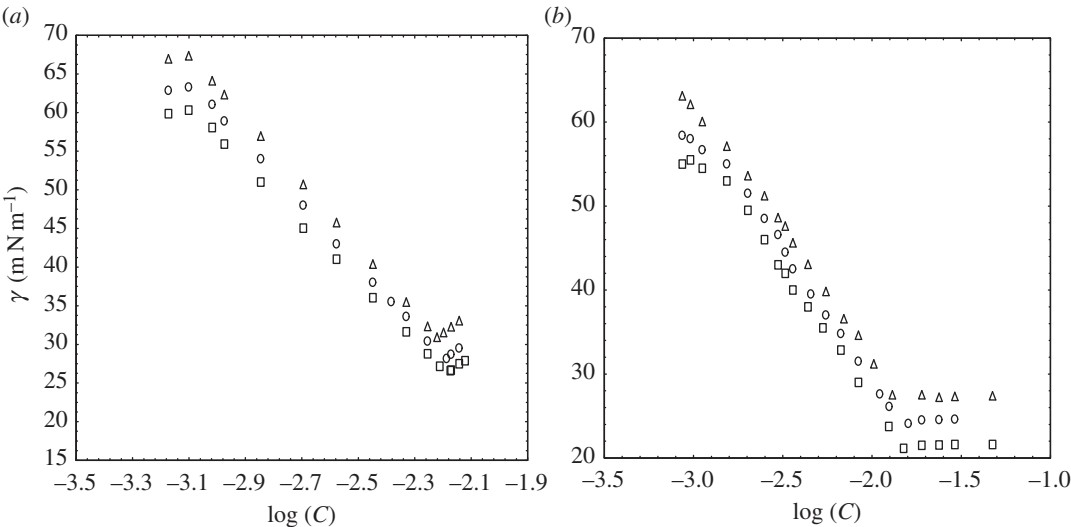

**Figure 1.** Plot of surface tension versus concentration of SDS-rich (*a*) and DTAB-rich (*b*) solution at $T = 293.15$ (triangle), 298.15 (circle) and 303.15 K (square).

for given DTAB-rich solution, and afterwards, a constant volume of aqueous MO or MB solution was added and the solution was mixed properly for 5 min. After that, the absorbance was noted.

By experimental data, the intensity of dye absorbance as a function of the concentration of DTAB-rich was examined. Using nonlinear regression procedure, MO and MB binding constant and MO and MB distribution constant between the aqueous phase and DTAB-rich micellar phase were calculated.

## 2.7. Zeta potential, polydispersity index and hydrodynamic radius measurements

Zeta potentials, polydispersity index (PDI) and hydrodynamic radius of DTAB-rich and SDS-rich surfactants in the presence and absence of MO and MB dye in the aqueous medium were measured by dynamic light scattering (DLS, MicrotracZetatrac, U2771). Using an aqueous surfactant, the set-zero was made to nullify their contribution in the formulations. The calibration was done using an aqueous dispersion of polystyrene in Milli-Q water for a standard particle diameter and zeta potential measurements at 298.15 K and $p = 0.1$ MPa with $\pm 2$ nm and $\pm 5$ mV uncertainties, respectively.

## 2.8. Krafft point measurements

The Krafft temperatures ($T_K$) of DTAB-rich and SDS-rich surfactant solutions were determined by placing the samples in a refrigerator at 278.15 K for at least 24 h, and the precipitation was observed. The temperature was raised slowly by recording the specific conductance after every 2 min till the steady value in the circulatory bath under constant stirring. The increment of temperature in the circulatory bath should be slow in order to make the solution homogeneous, and the mobility of the molecules of the mixed surfactants was regular, and the specific conductivity value was accurate without fluctuation in the experimental conductivity value. There was the temperature in which the specific conductance against temperature graph displayed the sudden alteration in the slope [33]. Such temperature was the same as that required for complete dissolution of the precipitated system into a clear solution. The Krafft temperature measurements reproducibility in each case was within $\pm 0.05$ K.

# 3. Results and discussion

## 3.1. Surface tension ($\gamma$) and surface properties

We can observe in figure 1*a* that the $\gamma$ initially decreases with an increment of the concentration of SDS-rich and goes to the lowest value which indicates the formation of the micelle and the break point is the CMC. Now for figure 1*b* of DTAB-rich, the $\gamma$ is reduced with the sharp break after which $\gamma$ stays very nearly constant. This meeting point provides CMC [34]. Table 2 contains the CMC of SDS-rich and DTAB-rich surfactant by tensiometry.

**Table 2.** Values of slope, critical micelle concentration (CMC), maximum surface excess concentration ($\Gamma_{max}$), area occupied by per surfactant molecule ($A_{min}$), surface pressure ($\pi_{cmc}$), packing parameter ($P$), standard free energy interfacial adsorption ($\Delta G^0_{ads}$) of SDS-rich and DTAB-rich in aqueous medium at $T = 293.15$, 298.15 and 303.15 K. The error limits of CMC, $\Gamma_{max}$, $A_{min}$, $\pi_{cmc}$, $P$ and $\Delta G^0_{ads}$ are within $\pm 3$, $\pm 5$, $\pm 4$, $\pm 3$, $\pm 4$ and $\pm 6\%$, respectively. Standard uncertainty is $u(T) = \pm 0.01$ K.

| temperature (K) | slope (mN m⁻¹ lnM⁻¹) | CMC (mol l⁻¹) | $\Gamma_{max}$ (10⁶ mol m⁻²) | $A_{min}$ (A°² molecule⁻¹) | $\pi_{cmc}$ (mNm⁻¹) | $P$ | $\Delta G^0_{ads}$ (kJ mol⁻¹) |
|---|---|---|---|---|---|---|---|
| SDS-rich | | | | | | | |
| 293.15 | −41.70 | 0.0060 | 3.71 | 44.70 | 41.95 | 0.47 | −43.39 |
| 298.15 | −38.40 | 0.0063 | 3.36 | 49.37 | 43.85 | 0.43 | −45.29 |
| 303.15 | −37.20 | 0.0067 | 3.20 | 51.82 | 44.61 | 0.41 | −46.31 |
| DTAB-rich | | | | | | | |
| 293.15 | −32.2 | 0.0135 | 2.87 | 57.89 | 45.35 | 0.36 | −49.5 |
| 298.15 | −31.8 | 0.01401 | 2.79 | 59.62 | 47.91 | 0.35 | −51.1 |
| 303.15 | −31.3 | 0.0147 | 2.70 | 61.59 | 49.99 | 0.34 | −52.5 |

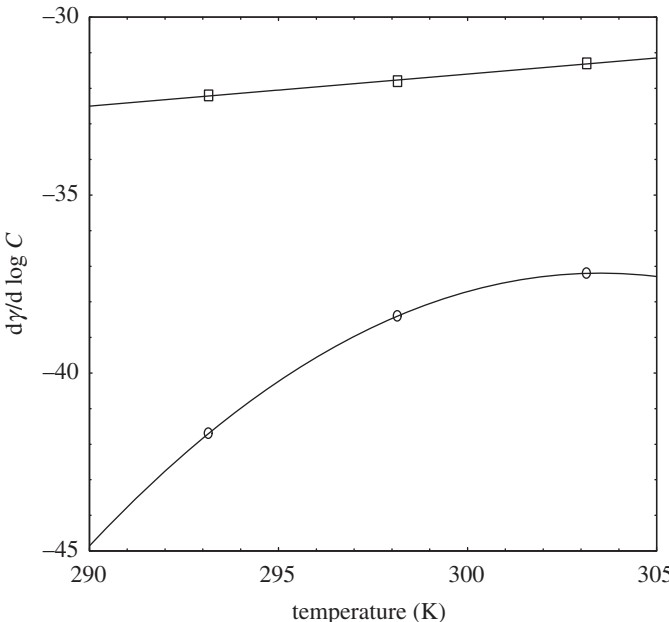

**Figure 2.** Variation of $(d\gamma/d\log C)$ with temperature (K) for DTAB-rich (square) and SDS-rich (circle).

The CMC of SDS-rich decreased to a value of 6.30 mM (table 2) in contrast with the CMC of 8.30 mM of SDS [35] at 298.15 K and the CMC of DTAB-rich diminished to a value of 14.01 mM (table 2) in contrast with the CMC of 14.6 mM of DTAB [36] at 298.15 K. The decrease in CMC was observed for cationic surfactant mixtures [37], due to the synergism in mixed micelle formation that is present when a mixture of two surfactants has a CMC lower than those of both individual surfactants [38]. The CMC increase here with an increase in temperature is due to the smaller probability of the hydrogen bond formation [39].

The slope $(d\gamma/d\log C)$ of the sigmoidal curve, where $C$ is a surfactant concentration (mol l$^{-1}$), gives numerous facts regarding the surface properties [34]. The plot of slope $(d\gamma/d\log C)$ of SDS-rich and DTAB-rich surfactant solutions with temperature is displayed in figure 2.

The slope can play a vital role in the surface properties of mixed surfactants (table 2). It is observed in figure 2, that both the graphs containing a minimal variation in nature. For DTAB-rich, the graph is best fitted with a polynomial of second degree fit, whereas, for SDS-rich, the graph is best fitted with a linear fit. From the slope $(d\gamma/d\log C)$ of the curve, the maximum surface excess concentration $(\Gamma_{\max})$ value is evaluated with Gibb's isotherm [34].

$$\Gamma_{\max} = -\frac{1}{2.303nRT}\left[\frac{d\gamma}{d\log C}\right]_{\mathrm{T,P}}. \tag{3.1}$$

Here, $R$ is value of gas constant and $n$ takes the value of 2 [36].

The surface excess concentration $(\Gamma_{\max})$ is an adequate measure of adsorption at the air/solution interface which depends upon the nature of surfactants.

The $\Gamma_{\max}$ of SDS-rich in water is observed as 3.71 µmol m$^{-2}$ (table 2) at 293.15 K which matched with the literature [22]. But there is a decrease in the value of $\Gamma_{\max}$ with an increase of temperature as 3.36 µmol m$^{-2}$ (table 2) at 298.15 K in contrast with the $\Gamma_{\max}$ of 2.65 µmol m$^{-2}$ of SDS in the aqueous system [35] at 298.15 K.

Similarly, $\Gamma_{\max}$ of DTAB-rich in water is noted as 2.87 µmol m$^{-2}$ (table 2) at 293.15 K which agreed with the reported study [22]. But there is a decrease in the value of $\Gamma_{\max}$ with an increase of temperature as 2.79 µmol m$^{-2}$ (table 2) at 298.15 K in contrast with the $\Gamma_{\max}$ of 1.56 µmol m$^{-2}$ of DTAB in the aqueous system [40] at 298.15 K.

Table 2 shows that the $\Gamma_{\max}$ values decrease with an increase in temperature which may be due to the enhanced molecular thermal agitation at higher temperature [41]. Such behaviours were also observed in the previous study [36].

The area occupied per surfactant molecule $(A_{\min})$ is calculated by following equation:

$$A_{\min} = \frac{1}{N\Gamma_{\max}}, \tag{3.2}$$

where $N$ stands for Avogadro's number.

Our data of $A_{min}$ value of SDS-rich in the water is 44.70 $A^{o2}$ molecule$^{-1}$ (table 2) at 293.15 K which is supported by the earlier result [22]. With increasing the temperature, the $A_{min}$ value is increased as 49.37 $A^{o2}$ molecule$^{-1}$ (table 2) at 298.15 K, while with anionic surfactant (SDS) in water $A_{min}$ value is 62.10 $A^{o2}$ molecule$^{-1}$ at 298.15 K [35].

In the same way, it is noted that the $A_{min}$ value of DTAB-rich in water as 57.89 $A^{o2}$ molecule$^{-1}$ (table 2) at 293.15 K is similar to the reported data [22]. With an increase in temperature, the $A_{min}$ value of DTAB-rich in water is observed as 59.62 $A^{o2}$ molecule$^{-1}$ (table 2) at 298.15 K, in contrast with the $A_{min}$ value of DTAB in water as 106.37 $A^{o2}$ molecule$^{-1}$ [40] at 298.15 K.

The increase in $A_{min}$ was observed with an increase in temperature process orientation as a result of the thermal molecular motion at high temperatures [42]. The $\Gamma_{max}$ and $A_{min}$ are inversely proportional with rising of the temperature. Such trends were also noted in earlier studies [36].

If there is no interaction [40] between SDS and DTAB, but they only compose the mixed adsorption film, the ideal area of the mixed adsorption film can be calculated as

$$A_{ideal} = \alpha, A_{min,1} + (1 - \alpha) A_{min,2}. \tag{3.3}$$

For SDS-rich, $\alpha = (0.75)$ is the mole fraction of the SDS in the total mixed solute. By substituting the values of $\alpha$, $A_{min,1}$ (62.1 $A^{o2}$ molecule$^{-1}$ and 63 $A^{o2}$ molecule$^{-1}$ for pure SDS) and $A_{min,2}$ (106.37 $A^{o2}$ molecule$^{-1}$ and 111 $A^{o2}$ molecule$^{-1}$ for pure DTAB) at 298.15 K and 303.15 K, respectively, in equation (3.3), we get $A_{ideal}$ (73.16 $A^{o2}$ molecule$^{-1}$ at 298.15 K and 75 $A^{o2}$ molecule$^{-1}$ at 303.15 K).

Similarly, For DTAB-rich, $\alpha = (0.75)$ is the mole fraction of the DTAB in the total mixed solute. By substituting the values of $\alpha$, $A_{min,1}$ (106.37 $A^{o2}$ molecule$^{-1}$ and 111 $A^{o2}$ molecule$^{-1}$ for pure DTAB) and $A_{min,2}$ (62.1 $A^{o2}$ molecule$^{-1}$ and 63 $A^{o2}$ molecule$^{-1}$ for pure SDS) at 298.15 K and 303.15 K, respectively, in equation (3.3), we get $A_{ideal}$ (95.30 $A^{o2}$ molecule$^{-1}$ at 298.15 K and 99 $A^{o2}$ molecule$^{-1}$ at 303.15 K).

It is observed that the values of $A_{ideal}$ are higher than $A_{min}$ of SDS-rich and DTAB-rich systems (table 2), and the values of $A_{ideal}$ are higher than $A_{min}$ of DTAB and SDS. Also, it is noted that the $A_{min}$ values for SDS-rich and DTAB-rich systems are lower in comparison with the pure SDS and DTAB. Such behaviour was noted in the literature [40]. The lowered $A_{min}$ values indicate a significant attractive interaction between the components of the mixed surfactant system and hence the strongest attraction between the oppositely charged head groups [43].

The surface pressure at the CMC ($\pi_{cmc}$) is calculated by using the following equation:

$$\pi_{cmc} = \gamma_o - \gamma_{cmc}, \tag{3.4}$$

where $\gamma_o$ and $\gamma_{cmc}$ have the usual meanings.

The calculated $\pi_{cmc}$ value of SDS-rich in water as 41.95 mN m$^{-1}$ (table 2) at 293.15 K closed with the literature [22]. With an increase in temperature, the $\pi_{cmc}$ value is also increased as 43.85 mN m$^{-1}$ (table 2) at 298.15 K, but there is the difference in $\pi_{cmc}$ of SDS in water as 32.43 mN m$^{-1}$ at 298.15 K [35].

Similarly, the $\pi_{cmc}$ value of DTAB-rich in water is found as 45.35 mN m$^{-1}$ (table 2) at 293.15 K and then closed with an already reported study [22]. With an increase in the temperature, the $\pi_{cmc}$ value is obtained as 47.91 mN m$^{-1}$ (table 2) at 298.15 K which is different from the $\pi_{cmc}$ value of DTAB in water as 29.35 mN m$^{-1}$ [40] at 298.15 K.

An increase in $\pi_{cmc}$ is observed with an increase in temperature. The reason is that $\pi_{cmc}$ is a measure of the efficiency of the surfactant to lower the surface tension of water. Such increase in $\pi_{cmc}$ with a rise in temperature was also noted in the earlier research work [44]. Normally, an increase in $\pi_{cmc}$ for mixed surfactants deals with an associative interaction [45]. Israelachvili *et al.* [46] have discussed the micellar shape. To find out the packing parameter, the surface area of amphiphiles in mixed micelles and micellar growth may be used as

$$P = \frac{V_o}{A_{min} l_c}. \tag{3.5}$$

The value of $P$ as 0.47 of SDS-rich in the aqueous system (table 2) at 293.15 K is similar to the literature data [22]. But with increase in the temperature, the $P$ value decreases as 0.43 (table 2) at 298.15 K in contrast with the value as 0.34 for $P$ of SDS in water at 298.15 K [35]. Similarly, $P$ of DTAB-rich in water is observed as 0.36 (table 2) at 293.15 K is close to the literature [22], and when temperature increases, there is a decrease in the $P$ value as 0.35 (table 2) at 298.15 K in contrast with the $P$ value as 0.20 at 298.15 K of DTAB in water [36].

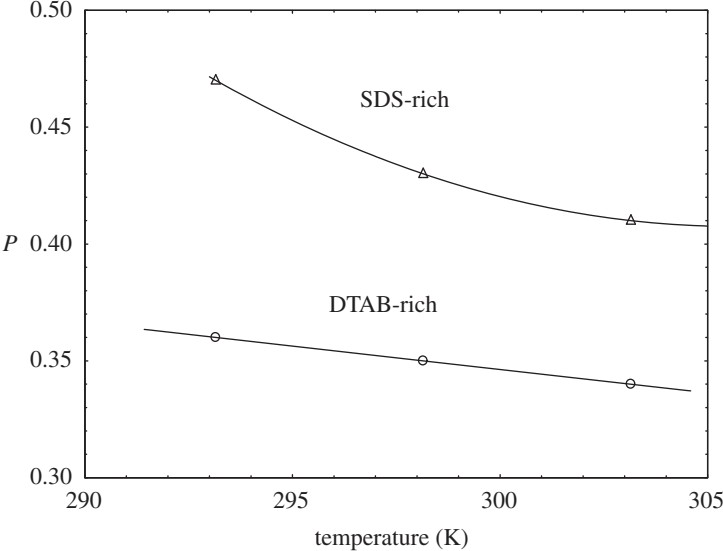

**Figure 3.** Variation of $P$ with temperature (K).

Thus, the packing parameter infers that the structural parameter of the surfactant at a molecular level with a favoured interfacial curvature is the resulting aggregate. The geometry of surfactant indicates that with the deep impact in the type of self-assembly structure of surfactants formed in solution, which is dependent on the packing parameter, amphiphiles can form different structures, spherical, cylindrical micelles, lamellar phases and reverse micelles, etc. [47].

Basically, the micelles are spherical for $p < 1/3$, cylindrical for $p < 1/2$ suggested by Israelachvili *et al.* [46]. In our investigation, from table 2, it is observed that $P$ values for both DTAB-rich and SDS-rich are higher than 0.3, indicating the formation of cylindrical or rod-shaped micelles. With an increasing temperature, the $P$ values decrease due to inducing the vibrational, rotational and translational oscillations with wreaking of binding forces of the SDS-rich and DTAB-rich mixed surfactants system [41]. Such a trend was also seen in the reported study [36]. The higher value of $P$ in mixed surfactants indicates disc-like micelles formation in water, which should also be suitable to form reverse micelles [48].

### 3.1.1. Correlation of packing parameter with temperature

The decrease of $P$ with temperature is sharply linear having a correlation coefficient ($r^2 = 1$) for DTAB-rich whereas SDS-rich shows the decrease of $P$ with temperature is in the concave pattern curve (figure 3).

From Tanford's formula [49], $V_0$ is the volume of exclusion per monomer in the micelle:

$V_0 = [27.4 + 26.9 n_c)] \text{Å}^3$, $l_c = [1.54 + 1.26(n_c)] \text{Å}$; $l_c$ is the highest chain length, whereas $n_c$ is carbon atoms in the chain of a hydrocarbon. The free energy of adsorption was calculated by using the following equation [46]:

$$\Delta G^o_{ads} = \Delta G^o_m - \frac{\pi_{cmc}}{\Gamma_{max}}. \tag{3.6}$$

From equations (3.1)–(3.6), $\Gamma_{max}$, $A_{min}$, $\pi_{cmc}$, $P$ and $G^o_{ads}$ are evaluated for DTAB-rich and SDS-rich and are displayed in table 2.

Our data of $\Delta G^o_{ads}$ of SDS-rich in water is observed as $-43.39 \text{ kJ mol}^{-1}$ (table 2) at 293.15 K and it is similar to the reported value [22], while at 298.15 K, the $\Delta G^o_{ads}$ value decreases as $-45.29 \text{ kJ mol}^{-1}$ (table 2) of SDS-rich system, and the value of $-51.80 \text{ kJ mol}^{-1}$ for $\Delta G^o_{ads}$ of SDS in aqueous system at 298.15 K was observed [40].

In the case of DTAB-rich in water, the $\Delta G^o_{ads}$ value is $-49.5 \text{ kJ mol}^{-1}$ at 293.15 K and then it is closed with the literature [22]. With an increasing temperature, the $\Delta G^o_{ads}$ value decreases as $-51.1 \text{ kJ mol}^{-1}$ (table 2) at 298.15 K in contrast with the value of $-48.63 \text{ kJ mol}^{-1}$ for $\Delta G^o_{ads}$ of DTAB in an aqueous system at 298.15 K [36].

Negative values of $\Delta G^o_{ads}$ indicate spontaneity of the adsorption of surfactant molecules on the surface. The $\Delta G^o_{ads}$ values become more negative on increasing the temperature, which is indicating

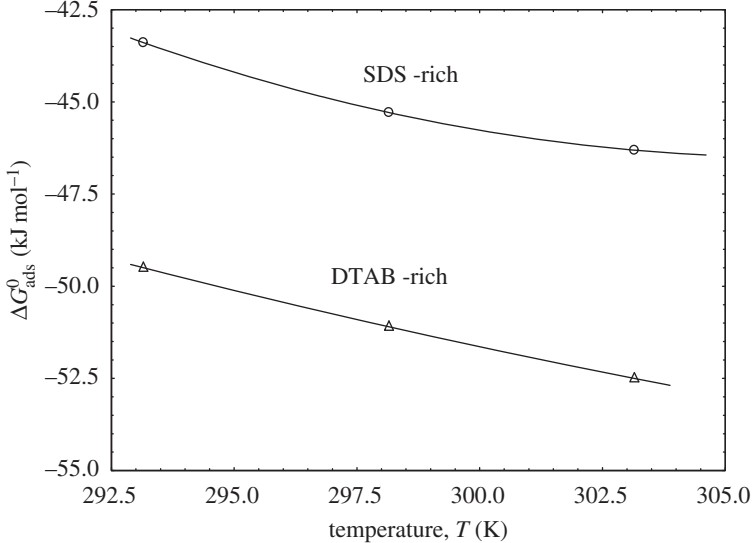

**Figure 4.** Variation of $\Delta G^o_{ads}$ with temperature (K).

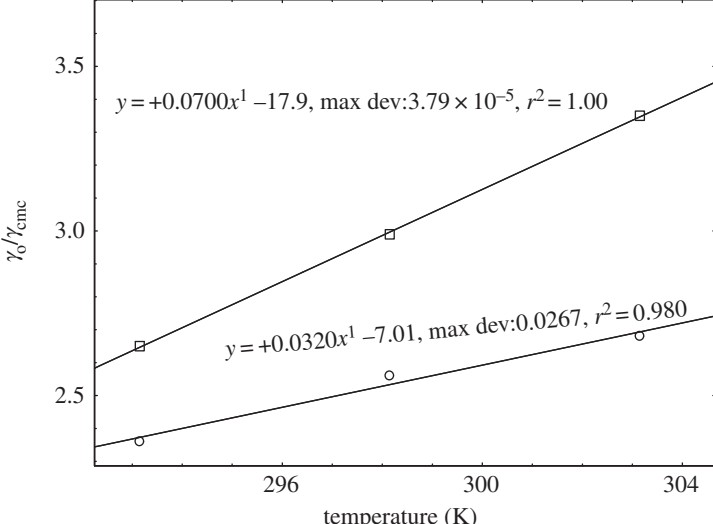

**Figure 5.** Variation of $\gamma_0/\gamma_{cmc}$ with temperature: open circles (SDS-rich) and open squares (DTAB-rich).

the higher spontaneity of adsorption of surfactant molecules on the surface; the negative $\Delta G^o_{ads}$ values were also observed in the literature [36].

### 3.1.2. Correlation of $\Delta G^o_{ads}$ with temperature

Gibbs energies of adsorption of SDS-rich and DTAB-rich show the unique variation. It is found that there is a concave curve with an increase of temperature for SDS-rich, whereas, in DTAB-rich, the curve became linear with a correlation coefficient ($r^2 = 1$) with an increase of temperature (figure 4).

### 3.1.3. Correlation of $\gamma_0/\gamma_{cmc}$ with temperature

Secondary parameters can be generated from the primary data of surface tension. In 2012, a new concept to describe the solvophobic effect [50] was proposed by Mukhim and Ismail. They calculated the ratio of the solvent surface tension to the solution surface tension at the CMC, $\gamma_0/\gamma_{cmc}$. This ratio can be used to describe the solvophobic effect [51]. Figure 5 shows that the variation of the ratio of the solvent surface tension to the solution surface tension at the CMC with temperature for DTAB-rich and SDS-rich systems. The plot of $\gamma_0/\gamma_{cmc}$ with the volume fraction of methanol at 293.15 K was noted for the concave nature of the curves by Pathak et al. [22] for DTAB-rich and SDS-rich systems, whereas we have seen the linear

**Table 3.** Values of solution surface tension ($\gamma_{cmc}$), solvent surface tension ($\gamma_0$), $\gamma_0/\gamma_{cmc}$, free energy of surface at equilibrium ($G_{min}$) and $\Delta G^o_{ads}/\Delta G^o_m$ of SDS-rich and DTAB-rich in aqueous medium at $T = 293.15$, 298.15 and 303.15 K. Errors limits of $\gamma_{cmc}$, $\gamma_0$ and $\Delta G_{min}$ are within $\pm$ 4%, $\pm$ 3% and $\pm$ 5%, respectively.

| temperature (K) | $\gamma_{cmc}$ (mN m$^{-1}$) | $\gamma_0$ (mN m$^{-1}$) | $\gamma_0/\gamma_{cmc}$ | $G_{min}$ (kJ mol$^{-1}$) | $\Delta G^o_{ads}/\Delta G^o_m$ |
|---|---|---|---|---|---|
| SDS-rich | | | | | |
| 293.15 | 30.80 | 72.75 | 2.362 | 8.29 | 1.35 |
| 298.15 | 28.16 | 72.01 | 2.557 | 8.37 | 1.40 |
| 303.15 | 26.60 | 71.21 | 2.677 | 8.30 | 1.43 |
| DTAB-rich | | | | | |
| 293.15 | 27.40 | 72.75 | 2.655 | 9.55 | 1.47 |
| 298.15 | 24.10 | 72.01 | 2.988 | 8.65 | 1.50 |
| 303.15 | 21.16 | 71.21 | 3.349 | 7.85 | 1.54 |

variation with temperatures for DTAB-rich and SDS-rich systems. But the DTAB-rich has the higher curve than SDS-rich with a slope (0.07) and the correlation coefficient ($r^2 = 1$) and whereas SDS-rich has a slope (0.032) and the correlation coefficient ($r^2 = 0.980$).

The observed $\Delta G^o_{ads}$ values are higher than $\Delta G^o_{min}$ values indicating the adsorption at the air–solution interface is more favourable than the formation of micelles in the bulk solution [52]. The ratio of $\Delta G^o_{ads}/\Delta G^o_m$ was found to be approximately 1.5 (table 3) for SDS-rich and DTAB-rich at different temperatures indicating less spontaneity in the transfer of the monomers to the interface [53].

Free energy of surface at equilibrium is also known as molar Gibbs energy at CMC for maximum adsorption attained. This is one of the thermodynamic parameters for the evolution of synergism in mixed adsorption film at equilibrium [54] and calculated as follows:

$$G_{min} = A_{min}\ \gamma_{cmc} N_A,\tag{3.7}$$

where $\gamma_{cmc}$ is the surface tension at CMC and $N_A$ is the Avogadro's number. We observed the lower values of $G_{min}$ in SDS-rich and DTAB-rich systems. The observed lower values of $G_{min}$ ascertain the thermodynamic stability [55]. Since the obtained $G_{min}$ values are lower in magnitude (table 3), it can be inferred that thermodynamically stable surfaces are formed with synergistic interaction [56].

As we have two systems (DTAB-rich and SDS-rich) and their interaction is discussed at three different temperatures above, different parameters from surface tension are studied. Here we want to take one representative system as DTAB-rich and compare with DTAB at 303.15 K by some parameters. Here, 21.16 is the $\gamma_{cmc}$ for DTAB-rich at 303.15 K (table 3), and 39 was the $\gamma_{cmc}$ of DTAB at 303.15 K [57]. The $\Gamma_{max}$ for DTAB-rich at 303.15 K is 2.70 (table 2), and 1.51 was the $\Gamma_{max}$ of DTAB at 303.15 K [57]. The $A_{min}$ for DTAB-rich at 303.15 K is 61.59 (table 2), and 111 was the $A_{min}$ of DTAB at 303.15 K [57]. The CMC for DTAB-rich at 303.15 K is 14.7 mM (table 2), and 15.3 was the CMC of DTAB at 303.15 K [57]. Thus we can say from the above comparison that the addition of DTAB into SDS leads to the strong electrostatic attraction between (+ve and −ve) charged head groups and the strengthening of the interaction between the SDS and DTAB molecules, thus leading to larger $\Gamma_{max}$, smaller $A_{min}$ and lower CMC and $\gamma_{cmc}$ at air/solution interface [58].

The further detailed investigation regarding the interaction between DTAB and SDS is also carried out in the following section by conductivity study.

## 3.2. Specific conductance measurement and thermodynamic properties

The specific conductance of the mixture of DTAB and SDS in the form of DTAB-rich and SDS-rich solutions in water for the calculation of the CMC at $T = 293.15$, 298.15 and 303.15 K are displayed in electronic supplementary material, figure S1. The specific conductance values of DTAB-rich and SDS-rich system increases with increment in temperature. The conductivity increases with increment in concentration with a certain slope. However, at a particular concentration, the slope changes for each plot. The break of two straight lines is indicated as the CMC. A degree of ionization ($\alpha$) can be obtained from the ratio of post-micellar ($S_2$) to the pre-micellar slope ($S_1$). The variations in pre- and post-micellar slopes on the plots of conductance with a concentration of the solution of DTAB-rich and SDS-rich solutions in the mixed surfactants are given in table 4. With an increase of temperature,

**Table 4.** Values of pre-micellar slope ($S_1$), post-micellar slope ($S_2$), degree of ionization ($\alpha$), critical micelle concentration (CMC), Gibb's free energy of micellization ($\Delta G_m^o$), standard enthalpy of micellization ($\Delta H_m^o$), the standard entropy of micellization ($\Delta S_m^o$) of SDS-rich and DTAB-rich in an aqueous medium at $T = 293.15$, 298.15 and 303.15 K. The error limits of CMC, $\alpha$, $\Delta G_m^o$, $\Delta H_m^o$ and $\Delta S_m^o$ are within $\pm 4$, $\pm 4$, $\pm 4$, $\pm 5$ and $\pm 5\%$, respectively. Standard uncertainty is $u(T) = \pm 0.01$ K.

| temperature (K) | $S_1$ (mS cm$^{-1}$ M$^{-1}$) | $S_2$ (mS cm$^{-1}$ M$^{-1}$) | $\alpha$ | CMC (mol l$^{-1}$) | $\Delta G_m^o$ (kJ mol$^{-1}$) | $\Delta H_m^o$ (kJ mol$^{-1}$) | $\Delta S_m^o$ (J mol$^{-1}$ K$^{-1}$) |
|---|---|---|---|---|---|---|---|
| SDS-rich | | | | | | | |
| 293.15 | 64.5 | 36.3 | 0.56 | 0.00592 | −32.096 | −9.876 | 75.7945 |
| 298.15 | 62.28 | 35.5 | 0.57 | 0.00620 | −32.25 | −10.040 | 74.4887 |
| 303.15 | 58.97 | 34.2 | 0.58 | 0.00650 | −32.38 | −10.459 | 72.3316 |
| DTAB-rich | | | | | | | |
| 293.15 | 105.0 | 35.8 | 0.34 | 0.01349 | −33.67 | −10.425 | 79.2864 |
| 298.15 | 100 | 35.0 | 0.35 | 0.01400 | −33.88 | −10.73 | 77.6400 |
| 303.15 | 94.4 | 34.1 | 0.36 | 0.01470 | −34.03 | −11.053 | 75.7985 |

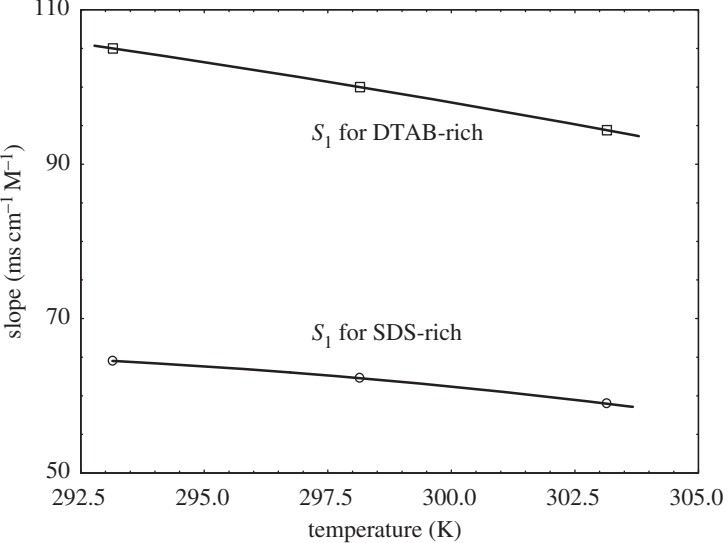

**Figure 6.** Variation of slope versus temperature in pre-micellar slope regions.

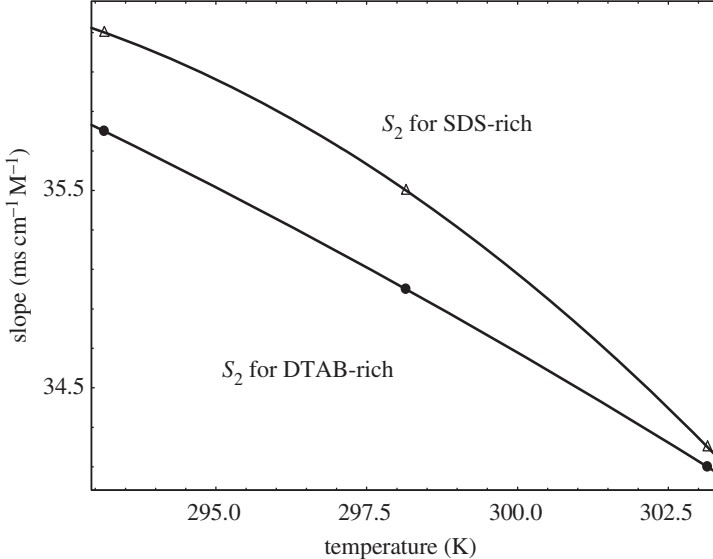

**Figure 7.** Variation of slope versus temperature in post-micellar slope regions.

both slopes (pre-CMC and post-CMC) decrease (table 4 and figures 6 and 7) for DTAB-rich and SDS-rich systems. But pre-micellar slopes decrease monotonously showing almost linear variation for DTAB-rich and SDS-rich systems and the post-micellar slopes decrease sharply showing almost linear variation for DTAB-rich, and the nature of the curve seems convex for SDS-rich system.

The mixed surfactants have shown the largest pre-CMC slopes and smallest post-CMC slopes leading to the lowest degrees of dissociation (table 4). These slopes are so sensitive, which decides the degrees of dissociation, CMC and other thermodynamic parameters.

The CMC obtained for DTAB-rich and SDS-rich system from conductivity measurements in water at three different temperatures are given in table 4. It indicates that the CMC increases with increasing temperature. The effect of temperature on micelle formation could be explained by two reasons. The first reason is as the temperature increases, the degree of hydration of the hydrophilic group decreases, which could favour micelle formation; however, a rise in the temperature also induces disruption of the water structure surrounding the hydrophobic group, and this is unfavourable to micelle formation. It seems from the data of table 4 that the second effect is a main studied temperature range [59].

The CMC of SDS-rich diminished to a value of 6.2 mM (table 4) in contrast with the CMC of 8.38 mM of SDS [60] at 298.15 K, and CMC of SDS-rich is observed as 6.5 mM (table 4) at 303.15 K in contrast with the CMC of 8.5 mM of SDS [61] at 303.15 K, while the CMC of DTAB-rich diminished to a value of

14.00 mM (table 4) in contrast with the CMC of 14.50 mM of DTAB [36] at 298.15 K and the CMC of DTAB-rich is observed as 14.70 mM (table 4) at 303.15 K in contrast with the CMC of 15.1 mM of DTAB [57] at 303.15 K by conductivity methods. Herrington *et al.* [62] observed the decrease in CMC for cationic surfactant mixtures. Such a decline in CMC happens in the more substantial synergistic effects between two oppositely charged surfactants that are mixed homogeneously.

### 3.2.1. Synergistic effects

The synergistic effects can be observed with the help of models. The synergism in the mixed surfactants is possible when the CMC of a mixture is less than that of individual surfactants among the mixture. So, we have selected our SDS-rich system for the study of synergistic effects because, in DTAB-rich system, the values of mixed CMCs are found to be lower than the CMC values of pure DTAB and higher than the CMC values of pure SDS. Moreover, we have tested the values for $\beta^m$ (the micellar interaction parameter) of DTAB-rich and SDS-rich systems from the conductivity study with the famous equations given in the literature [38] as

$$x^2 \ln\left(\alpha\frac{\text{CMC}}{x\text{CMC}_1}\right)\bigg/(1-x)^2\ln\left[\frac{(1-\alpha)\text{CMC}}{(1-x)\text{CMC}_2}\right] = 1 \qquad (3.8)$$

and

$$\beta^m = \ln\left(\alpha\frac{\text{CMC}}{x\text{CMC}_1}\right)\bigg/(1-x)^2. \qquad (3.9)$$

For SDS-rich, CMC = CMC of mixed surfactant system for SDS-rich system, $\text{CMC}_1$ = CMC of pure SDS for SDS-rich system and $\text{CMC}_2$ = CMC of pure DTAB for SDS-rich system. The $\alpha$ and $x$ are the mole fractions of the SDS in the total mixed solute and the mixed micelles, respectively.

Similarly, for DTAB-rich, CMC = CMC of mixed surfactant system for DTAB-rich system, $\text{CMC}_1$ = CMC of pure DTAB for DTAB-rich system and $\text{CMC}_2$ = CMC of pure SDS for DTAB-rich system. The $\alpha$ and $x$ are the mole fractions of the DTAB in the total mixed solute and in the mixed micelles, respectively.

We found that $\beta^m$ was positive for DTAB-rich system as 0.5677 at 293.15 K, 0.8077 at 298.15 K and 0.8284 K at 303.15 K, by solving equation (3.8) iteratively to obtain the value of $x$ and then putting the value of $x$ into equation (3.9), whereas the negative values of $\beta^m$ were found for SDS-rich system (table 5). The evaluated positive and negative values of $\beta^m$ for DTAB-rich and SDS-rich systems were used in Rubingh's equations (3.10) and (3.11) to calculate the activity coefficients ($\gamma_1$ and $\gamma_2$). DTAB-rich systems generated values greater than 1 for $\gamma_1$ and $\gamma_2$, whereas SDS-rich systems produced values less than 1 for $\gamma_1$ and $\gamma_2$.

$$ln\gamma_1 = \beta^m(1-x)^2 \qquad (3.10)$$

and

$$ln\gamma_2 = \beta^m x^2. \qquad (3.11)$$

As the values of $\gamma_1$ as well as $\gamma_2$ in aqueous as well as other media in the entire study, $\alpha$ are less than unity, showing the synergistic interactions as well as non-ideal behaviour of the mixed systems [63].

So, DTAB-rich system explains the antagonistic interaction whereas SDS-rich system explains synergistic interaction. Therefore, we proceeded with further investigations on SDS-rich system.

The obtained values of $\gamma_1$ and $\gamma_2$ were less than 1, so the Clint condition of ideality was not recovered. So, we used our data to calculate the mixed CMC from the equation

$$\frac{1}{\text{CMC}} = \frac{\alpha}{\gamma_1\ \text{CMC}_1} + \frac{1-\alpha}{\gamma_2\ \text{CMC}_2}. \qquad (3.12)$$

Thus obtained mixed CMC values were lower than the measured experimental CMC of our system (tables 4 and 5) even though $\beta^m$ were the negative values (table 5). So, we used the pseudo phase separation model coupled with the dissociated Margules model [21] and obtained the almost closer mixed CMC in comparison with the experimental mixed CMC (tables 4 and 5 and figure 8).

The interesting facts are that the Margules model used equations (3.13) and (3.14) if $A_{12} = A_{21} = \beta^m$, then equations (3.13) and (3.14) recover to equations (3.10) and (3.11).

$$ln\gamma_1 = [A_{12} + 2(A_{21} - A_{12})\,x](1-x)^2 \qquad (3.13)$$

and

$$ln\gamma_2 = [A_{21} + 2(A_{12} - A_{21})\,(1-x)]x^2. \qquad (3.14)$$

**Figure 8.** Variation of CMC versus temperature: closed circles: calculated CMC values from Rubingh model; open circles: experimental CMC values from the conductivity study; dashed lines are the CMC values from the dissociated Margules model.

**Table 5.** Values of $\beta^m$, critical micelle concentration (CMC) from Rubingh model and dissociated Margules model, $A_{12}$ and $A_{21}$ of SDS-rich in aqueous medium at $T = 293.15$, 298.15 and 303.15 K.

| temperature (K) | $\beta^m$ | CMC from Rubingh model (mM) | CMC from Margules model (mM) | $A_{12}$ | $A_{21}$ | CMC of SDS CMC$_1$ (mM) | CMC of DTAB CMC$_2$ (mM) |
|---|---|---|---|---|---|---|---|
| 293.15 | $-2.727$ | 4.61 | 5.96 | $-1.70$ | $-3.897$ | 8.03[a] [60] | 15.38[a] [64] |
| 298.15 | $-2.450$ | 5.07 | 6.20 | $-1.65$ | $-3.088$ | 8.38[a] [60] | 14.50[a] [36] |
| 303.15 | $-2.436$ | 5.16 | 6.50 | $-1.51$ | $-3.681$ | 8.5[a] [61] | 15.1[a] [57] |

[a]Conductivity methods.

By using possible dissociation in SDS-rich system, then equations (3.13) and (3.14) can be changed into

$$ln\gamma_{1,r_1} = [\, A_{12} + 2(A_{21} - A_{12})\, x_{,r_1}]\, (1 - x_{,r_1})^2 \qquad (3.15)$$

and

$$ln\gamma_{1,r_2} = [\, A_{21} + 2(A_{12} - A_{21})\, (1 - x_{,r_1})]x_{,r_1}^2. \qquad (3.16)$$

The details about these two parameters ($A_{12}$ and $A_{21}$) as well as $\gamma_{1,r_1}$, $\gamma_{1,r_2}$ and $x_{,r_1}$ were described in the literature [21]. Then for the above SDS-rich systems, $A_{12}$ and $A_{21}$ were iteratively evaluated with the help of equations (3.13)–(3.16) and are given in table 5.

The value of $(r_1A_{12} + r_2A_{21})/(r_1 + r_2)$ is equivalent to $\beta^m$ when cationic and ionic surfactant mixed at equimolar composition and $\beta^m$ acts as a measure of the excess interaction between the two different surfactants in mixed micelles. Here, we suppose that surfactant SDS generates $r_1$ particles, and surfactant DTAB generates $r_2$ particles. It was found in the literature [21] that for the near-symmetric DTAB/SDS/H$_2$O mixed system, the two surfactants DTAB and SDS had the same alkyl chain and similar surface activities, their interaction was useful for the complete dissociation of surfactants, both $r_1$ and $r_2$ were chosen as 2 and observed more negative values of $A_{12}$ and $A_{21}$; whereas, in our SDS-rich system, we have used $r_1$ and $r_2$ as 1 instead of 2, because $r_1$ and $r_2$ as 2 generated the positive values of $A_{12}$ and $A_{21}$ which is in contrast with synergistic effects.

By using $r_1 = r_2 = 1$ in our system, we get the value of $A_{12}$ and $A_{21}$ less negative in comparison with the literature [21]. That may be one of the reasons we observed different values of $A_{12}$ and $A_{21}$, and other reason may be that we used asymmetrical amounts of SDS and DTAB. Hao et al. [21] used almost the symmetrical amount of SDS and DTAB, which interact highly by generating more negative values of

$A_{12}$ and $A_{21}$, whereas our SDS-rich system gives less negative values of $A_{12}$ and $A_{21}$ as SDS is more in excess in the mixture, which lowers the value of $r_1$ and $r_2$. Such types of behaviour were also discussed in the literature [65]. When the deviation is more, the degree of interaction between the mixed systems is higher. Hence the positive deviation as in our DTAB-rich system explains the antagonistic interaction, while the negative deviation as in our SDS-rich indicates the synergistic interaction. Therefore, there is still a need to have a huge research scope for the mixed micellization of opposite charged surfactants [66].

The free energies of micelle formation are calculated by a pseudo-phase separation model [22] and given in table 4.

$$\Delta G_m^o = (2 - \alpha)RT \, lnX_{cmc}, \tag{3.17}$$

where $X_{cmc}$, $R$ and $T$ have the usual meanings.

The value of $\alpha$ for SDS-rich in water is noted to be 0.56 (table 4) at 293.15 K and closed with the literature [22]. When the temperature has increased, the value of $\alpha$ is increased as 0.57 (table 4) at 298.15 K, but $\alpha$ of SDS in water was noted as 0.45 [67] at 298.15 K.

In the same way, the $\alpha$ value for DTAB-rich in water is noted as 0.34 (table 4) at 293.15 K and similar to the previously reported study [22]. But $\alpha$ of DTAB-rich in water is observed as 0.35 (table 4) at 298.15 K whereas $\alpha$ of DTAB in water as 0.21 was noted in the literature [36] at 298.15 K, and $\alpha$ of DTAB-rich is observed as 0.36 (table 4) at 303.15 K in contrast with the $\alpha$ of 0.29 of DTAB [57] at 303.15 K.

The increase in $\alpha$ with temperature is because of the decreasing charge density on the micellar surface. A more significant fraction of the counterions is dissociated and increasing the temperature reduces the aggregation number of the ionic surfactants [68].

The data of $\Delta G_m^o$ for SDS-rich in water is found as $-32.096$ kJ mol$^{-1}$ (table 4) at 293.15 K, which is similar to the earlier studied work [22], and an increase of temperature shows the free energy formation is more negative as $-32.25$ kJ mol$^{-1}$ (table 4) at 298.15 K in contrast with the $\Delta G_m^o$ value for SDS in water as $-34.04$ kJ mol$^{-1}$ [67] at $T = 298.15$ K, and $\Delta G_m^o$ of SDS-rich is observed as $-32.38$ kJ mol$^{-1}$ (table 4) at 303.15 K in contrast with the $\Delta G_m^o$ of $-36.2$ kJ mol$^{-1}$ of SDS [61] at 303.15 K.

In the case of DTAB-rich in water, $\Delta G_m^o$ is $-33.67$ kJ mol$^{-1}$ (table 4) at 293.15 K and a similar result was noted in the earlier study [22], and an increase of temperature shows the free energy formation is more negative as $-33.88$ kJ mol$^{-1}$ (table 4) at 298.15 K in contrast with the $\Delta G_m^o$ value for DTAB in water as $-35.25$ kJ mol$^{-1}$ [69] at 298.15 K, and $\Delta G_m^o$ of DTAB-rich is observed as $-34.03$ kJ mol$^{-1}$ at 303.15 K (table 4) in contrast with the $\Delta G_m^o$ of $-35.73$ kJ mol$^{-1}$ of DTAB [69] at 303.15 K.

The $\Delta G_m^o$ is negative with all systems and becomes more negative as the temperature increases (table 4). The higher negative $\Delta G_m^o$ with increasing temperature indicates that the micellization process is spontaneous and becomes more spontaneous with an increase in temperature. Our results are also supported by the literature [70]. The decreasing value of $\Delta G_m^o$ is attributed to the tendency to drive equilibrium towards hydrophobic bonding as temperature increased. Furthermore, the free energy of micelle formation is more negative in DTAB-rich systems in contrast to the SDS-rich systems at investigated temperatures whereas the free energy of micelle formation is less negative in DTAB-rich and SDS-rich systems in comparison to pure DTAB and SDS at investigated temperatures. Less negative $\Delta G_m^o$ indicates co-solute does not facilitate the micellization.

Standard enthalpies of micelle formation, the $\Delta H_m^o$ value is calculated by the Gibbs–Helmholtz equation as

$$\Delta H_m^o = -RT^2(2 - \alpha)\left[\frac{\partial lnX_{cmc}}{\partial T}\right]_P \tag{3.18}$$

by fitting the graph of $lnX_{cmc}$ against temperature, and then the term $\left[\frac{\partial lnX_{cmc}}{\partial T}\right]_P$ is calculated. $\Delta S_m^o$ can be evaluated with the help of $\Delta G_m^o$ and $\Delta H_m^o$ by the following equation (3.19):

$$T\Delta S_m^o = \Delta H_m^o - \Delta G_m^o. \tag{3.19}$$

Our data of $\Delta H_m^o$ for SDS-rich in water is obtained as $-9.876$ kJ mol$^{-1}$ (table 4) at 293.15 K and increase of temperature shows $\Delta H_m^o$ is more negative as $-10.04$ kJ mol$^{-1}$ (table 4) at 298.15 K while the $\Delta H_m^o$ value for SDS in water as $-14.49$ kJ mol$^{-1}$ was noted in the literature [67] at $T = 298.15$ K. In the case of DTAB-rich in water, $\Delta H_m^o$ is $-10.425$ kJ mol$^{-1}$ (table 4) at 293.15 K and increase of temperature shows $\Delta H_m^o$ is more negative as $-10.73$ kJ mol$^{-1}$ (table 4) at 298.15 K whereas the value of $\Delta H_m^o$ for DTAB in water was $-9.08$ kJ mol$^{-1}$ [69] at 298.15 K, and $\Delta H_m^o$ of DTAB-rich is observed as $-11.053$ kJ mol$^{-1}$ (table 4) at 303.15 K in contrast with the $\Delta H_m^o$ of $-9.38$ kJ mol$^{-1}$ of DTAB [69] at 303.15 K.

Table 4 shows that the micellization is exothermic at $T = 293.15$, 298.15 and 303.15 K for the DTAB-rich and SDS-rich system. Hence, $\Delta H^o_m$ is negative with all systems and becomes more negative as the temperature increases (table 4). It depicts the decrease in energy required for breaking the iceberg structure surrounding the alkyl chains of DTAB-rich and SDS-rich. Also, $\Delta H^o_m$ is more negative in DTAB-rich systems in comparison with pure DTAB, whereas in SDS-rich systems at 298.15 K, $\Delta H^o_m$ is less negative in comparison with pure SDS. Less negative value of $\Delta H^o_m$ may be due to an increase of hydrophobic interaction.

In our study of DTAB-rich and SDS-rich system, DTAB-rich are highly exothermic, primarily because of tail association [71]. Negative enthalpy values infer the importance of London dispersion interactions (LDI) as an attractive force for micellization for DTAB-rich and SDS-rich system [72]. The negative values of enthalpies of micellization for DTAB/SDS mixtures at 303.15 K found from microcalorimetry [73] also support our findings of enthalpies of micellization.

Our calculated data of $\Delta S^o_m$ for SDS-rich in water is found as 75.7945 J mol$^{-1}$ K$^{-1}$ (table 4) at 293.15 K and decreases with increasing the temperature as 74.4887 J mol$^{-1}$ K$^{-1}$ (table 4) at 298.15 K, whereas the value of $\Delta S^o_m$ for SDS in water as 65.67 J mol$^{-1}$ K$^{-1}$ was mentioned [67] at 298.15 K. With DTAB-rich in water, $\Delta S^o_m$ is 79.2864 J mol$^{-1}$ K$^{-1}$ (table 4) at 293.15 K, and increase of temperature shows $\Delta S^o_m$ decreases as 77.6400 J mol$^{-1}$ K$^{-1}$ (table 4) at 298.15 K, while the $\Delta S^o_m$ value for DTAB in water as 87.8 J mol$^{-1}$ K$^{-1}$ [69] at 298.15 K and $\Delta S^o_m$ of DTAB-rich is observed as 75.7985 J mol$^{-1}$ K$^{-1}$ (table 4) at 303.15 K, in contrast with the $\Delta S^o_m$ of 87 J mol$^{-1}$ K$^{-1}$ of DTAB [69] at 303.15 K. Moreover, the values of $\Delta S^o_m$ in DTAB-rich solutions were observed to be high, in contrast to the SDS-rich solutions at investigated temperatures.

On increasing the temperature, the $\Delta S^o_m$ value decreases for DTAB-rich and SDS-rich system. It may be because of the disruption of the iceberg water structure around the alkyl group with increasing the kinetic energy of the system.

In our study, we observed the positive value of $\Delta S^o_m$ with DTAB-rich and SDS-rich systems. It infers that a liquid–phase aggregate could be formed, whereas a negative value of $\Delta S^o_m$ may indicate the formation of the solid–phase aggregate [71].

Therefore, the negative values of $\Delta H^o_m$ and $\Delta G^o_m$ and positive values of $\Delta S^o_m$ are indicative for DTAB and SDS interactions. Such behaviour is also noted in the earlier study [74]. But in the literature [71], the negative values of entropy have been reported.

Electronic supplementary material, figure S2 shows the relationship of enthalpy–entropy compensation phenomenon for DTAB-rich and SDS-rich in the water at $T = 293.15$, 298.15 and 303.15 K. A linear relationship is obtained for $\Delta H^o_m - \Delta S^o_m$ and is expressed with the help of the following equation:

$$\Delta S^o_m = \frac{1}{T_c}\Delta H^o_m + \sigma, \tag{3.20}$$

where $1/T_c$ is slope and $\sigma$ are intercepts of a linear plot. $T_c$ measures solvation part of micellization while $\sigma$ determines the solute–solvent interaction. $T_c$ values obtained in water for DTAB-rich and SDS-rich are 179.86 K and 172.71 K, respectively. $\Delta G^o_m$, $\Delta H^o_m$ and $\Delta S^o_m$ of DTAB-rich and SDS-rich in water at $T = 293.15$, 298.15 and 303.15 K are displayed in table 4, whereas the values of $T_c$, $\sigma$ and $\Delta_m C^o_p$ for DTAB-rich and SDS-rich solutions by using conductivity measurement are given in table 6.

The heat capacity of micellization ($\Delta_m C^o_p$) which is obtained from the slope of $\Delta H^o_m$ against temperature curve (electronic supplementary material, figure S3) is noted as

$$\Delta_m C^o_p = \frac{\partial \Delta_m H^o}{\partial T}. \tag{3.21}$$

The negative value of the heat capacity of micellization ($\Delta_m C^o_p$) is observed for the self-association of amphiphiles.

## 3.3. UV–visible absorption spectroscopy

### 3.3.1. Optical analysis

Systematic UV–Vis spectroscopic investigations of DTAB-rich and SDS-rich solutions were carried out to understand the behaviour of mixed surfactants association in water. The absorption maxima of an aqueous solution of DTAB-rich were found at 210 nm and 280 nm, respectively (figure 9).

With increasing the concentration of DTAB into aq-SDS initially, the absorbance intensity is increased significantly due to the hyperchromic shift as for 0.005 M DTAB only; the maximum absorbance of 2.377

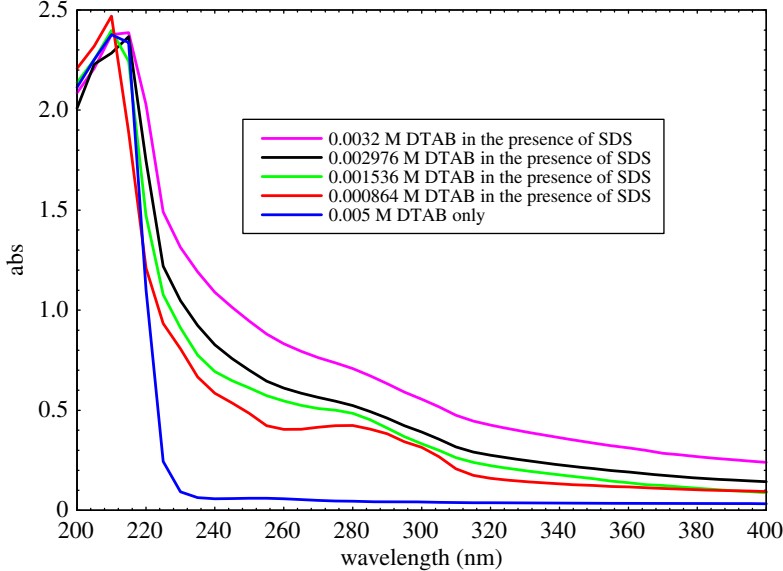

**Figure 9.** Absorption spectra of DTAB-rich surfactant.

**Table 6.** Compensation parameters: $T_c$, $\sigma$ and $\Delta_m C_p^0$ value of SDS-rich and DTAB-rich by using conductivity measurement.

| compensation parameters | SDS-rich | DTAB-rich |
| --- | --- | --- |
| $T_c$(K) | 172.71 | 179.86 |
| $\sigma$ | 132.8 | 137.2 |
| $\Delta_m C_p^o$ (J mol$^{-1}$ K$^{-1}$) | −58.2 | −62.8 |

at 210 nm is noted. When adding the constant amount of 0.01 M SDS in 0.000864 M DTAB, the absorbance increases up to 2.469 without red or blue shift, but only the hyperchromic shift is observed. When the amount of DTAB increases up to 0.001536 M in the constant concentration of 0.01 M SDS, i.e. the absorbance decreases as 2.398, so there is the observation of hypochromic shift. Further, adding more DTAB as 0.002976 M with 0.01 M SDS, there is the observation of hypochromic shift as well as red shift containing absorbance as 2.367 at 215 nm, and again on increasing the concentration of DTAB as 0.0032 M with 0.01 M SDS, there is the observation of hyperchromic shift containing absorbance as 2.387 at 215 nm, while at 280 nm the hyperchromic shift is observed with increasing DTAB concentration due to the weakening of binding forces, van der Waals forces and electrostatic interaction. The results depicted that the hydrophobic interactions among surfactant tails also give rise to the higher adsorption of surfactant molecules. Such behaviour was also observed for cationic-rich and anionic-rich mixtures of CTAB and SDS in the literature [75].

The absorption maxima of an aqueous solution of SDS-rich were found at 270 nm and 360 nm, respectively (figure 10).

With increasing the concentration of SDS into aq-DTAB initially, the absorbance intensity is increased significantly due to the hyperchromic shift as for 0.01 M SDS only; the maximum absorbance of 0.039 at 270 nm is noted. When adding the constant amount of 0.005 M DTAB into 0.000096 M SDS, the absorbance increases up to 0.248 without red or blue shift but only hyperchromic shift is observed. When the amount of SDS increases up to 0.00048 M in the constant amount of 0.005 M DTAB, i.e. the absorbance also increases as 0.424, the observation is a hyperchromic shift. On further adding more SDS as 0.000672 M to 0.00096 M with 0.005 M DTAB, there is the observation of hyperchromic shift for both cases as 0.597 and 0.768 absorbances. In the same manner, for 360 nm, there is also absorbance of 0.01 for 0.01 M SDS only. When the constant DTAB is added as 0.005 M in 0.000096 M SDS, there is the progressive evolution of the absorbance bands as 0.169 at 360 nm. In the case of 0.00048 M, 0.000672 M and 0.00096 M SDS, the constant amount of 0.005 M DTAB interacts with SDS solutions to give the absorbance of 0.346, 0.401 and 0.449, respectively. Hence, two visible peaks at 270 and 360 nm of SDS-rich surfactant mixture give only the hyperchromic shift, but not visible blue

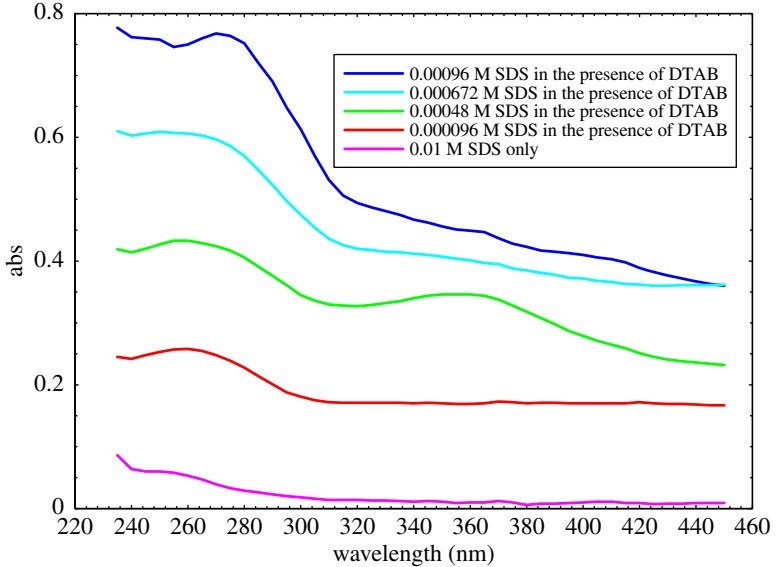

**Figure 10.** Absorption spectra of SDS-rich surfactant.

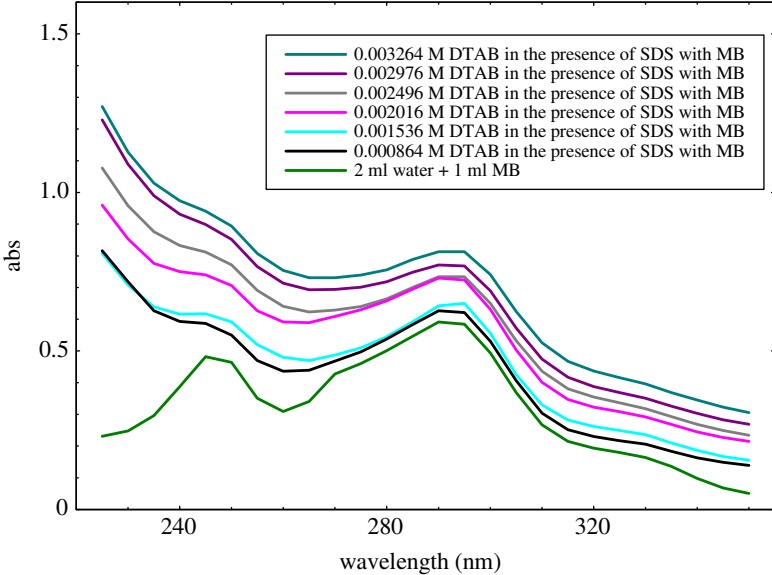

**Figure 11.** Absorption spectra of DTAB-rich surfactant in the presence of MB.

or red shift is noted at the investigated concentrations of SDS solution in the presence of DTAB. Thus, our results show that the hydrophobic interactions among surfactant tails give rise to the higher adsorption of surfactants molecules. Liu *et al.* have also been observed such type of behaviour of variable concentration of SDS in the aqueous medium [76]. Our findings of SDS-rich surfactant mixtures are also supported from the literature [24].

### 3.3.2. Interaction between dyes and mixed surfactants

Figure 11 shows that the absorbance intensity of the MB in aqueous solution affected with increasing the DTAB concentration. Thus, the MB and DTAB have positive charges while with SDS is opposite so they could be induced to have a weak interaction with the hyperchromic shift. With increasing the concentration of surfactants, the maximum MB molecules could be accommodated into normal micelles as monomeric molecules, and absorbance intensity sharply increases with the hyperchromic shift [77].

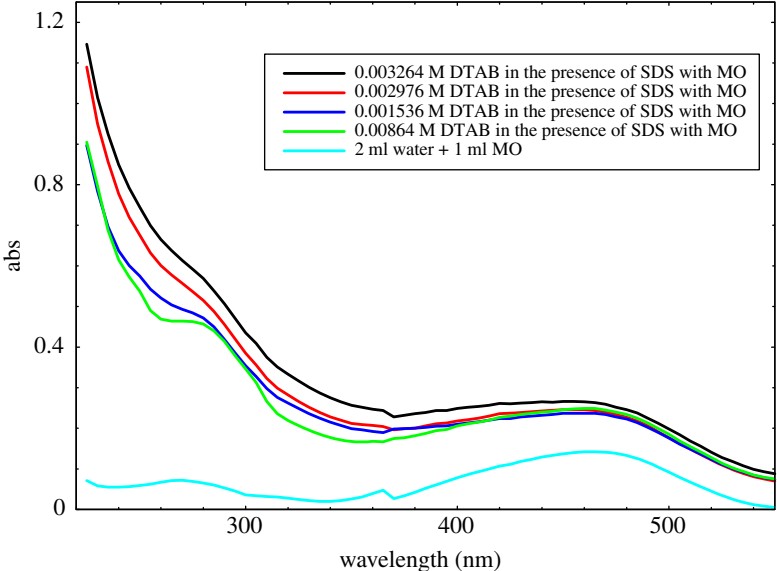

**Figure 12.** Absorption spectra of DTAB-rich surfactant in the presence of MO.

Figure 12 shows that the absorption peaks are obtained from MO in the aqueous medium. Such characteristic of the MO could be used for studying the DTAB-rich surfactants aggregates in the aqueous medium. The influences of different concentration of DTAB on the absorption spectrum of MO were studied by using UV–Vis spectroscopy. On increasing the concentration of DTAB, the absorbance intensity increases monotonously with the hyperchromic shift, which could induce strong interaction between MO and DTAB-rich solution as compared to MB with DTAB-rich solution. Our patterns of spectra are also supported by the reported study [78]. In the case of DTAB-rich with MB and MO solution showing more hydrophobic micellar environment, the maximum absorbance of dyes with surfactants is observed. In the case of SDS-rich interaction with MB and MO solution, there was cloudy in the preparation of the solutions. Two intense absorption bands were registered between 275 and 465 nm for MO, and between 245 and 295 nm for MB, which is in good agreement with the literature data [79,80], as shown in figures 11 and 12.

Moreover, there is a progressive evolution of these bands with an increase of DTAB-rich concentration. The interaction of MO and MB with DTAB-rich has non-visibly shifted (figures 11 and 12). By assuming the ideal behaviour of dye in both phases, the distribution constant of dye following Nernstian distribution law can be written as

$$K = \frac{m_m}{m_o}. \tag{3.22}$$

Here, $K$ is distribution constant, whereas $m_m$ and $m_o$ are the concentrations of dye in micellar and aqueous phases. In the analysis of spectra of UV–Vis, parameter $A$ was calculated. Following Magid et al. [81], the ratio of dye absorbance's band intensities ($A$) was described as a function of cationic rich surfactant concentration and $A$ value is a sum of two parameters,

$$A = x_m A_m + (1 - x_m)A_o, \tag{3.23}$$

where $x_m$ is the dye mole fraction, $A_m$ is the value of $A$ corresponding to the situation when dye completely dissolves in micellar phases, whereas $A_o$ related to aqueous phases. In our case, dyes dissolve in pure water so the distribution constant can be presented as

$$K = \frac{n_m}{n_0[\text{DTAB} - \text{SDS}]M_{\text{DTAB}-\text{SDS}}}, \tag{3.24}$$

where $n_m$ is the moles of dye in aqueous phase whereas $n_0$ is with micellar phases, the [DTAB-SDS] is the molarity of DTAB in the presence of SDS, and the $M_{\text{DTAB}-\text{SDS}}$ is molecular weight of DTAB in the presence of SDS. Following the procedure developed in the literature [81] on analysis of the UV–Vis

**Table 7.** The fitting parameters ($A_m$ and $K'$) obtained in the nonlinear regression procedure (NLREG) of the UV–Vis data for dyes partitioning between water and (DTAB-SDS) micelles measured at room temperature and $K$ (distribution constant) calculated from relation: ($K' = K * M_{DTAB-SDS}$).

| S.N. | $A_m^a$ | $K'$ (binding constant) [l mol$^{-1}$] | $K$ (distrib. constant) |
|---|---|---|---|
| | methyl orange (MO) | | |
| 1. | 0.4290 | 21200 | 68753.48 |
| | methylene blue (MB) | | |
| 2. | 0.0116 | 126 | 408.6300 |

$^a$Absorbance of dyes dissolved in micellar phase.

spectra, the binding constant $K'$ of dye to reversed micelle is given as

$$K' = \frac{n_m}{n_0[\text{DTAB} - \text{SDS}]}. \tag{3.25}$$

There is a connection between $K$ and $K'$ as $K' = K * M_{DTAB-SDS}$.

According to Poisson distribution [59], equation (3.25) can be written as

$$x_m = \frac{K'[\text{DTAB} - \text{SDS}]}{1 + K'[\text{DTAB} - \text{SDS}]}. \tag{3.26}$$

Combining equations (3.23) and (3.26), we get

$$A = A_o + \frac{(A_m - A_o)K'[\text{DTAB} - \text{SDS}]}{1 + K'[\text{DTAB} - \text{SDS}]}. \tag{3.27}$$

Here $A_m$ and $K'$ were fitted parameters, $A_o$ values were found in a separate experiment. Obtained results are presented in electronic supplementary material, figures S4 and S5. Fitted parameters and constant distribution values are summarized in table 7. Table 7 indicates that the stronger interactions between MO with DTAB-rich micelles appeared in comparison among MB with DTAB-rich micelles.

## 3.4. Zeta potential, polydispersity index and hydrodynamic radius measurement for stability analysis

The $\xi$ is a potential that exists among the particle surface and dispersing liquid which changes according to the distance from the particle surface. The greater the positive/negative charge of the $\xi$, the more stable the particles are (due to electrical repulsion). Nevertheless, the higher $\xi$ value describes the higher dispersed condition of the particle in an aqueous media. For an adsorbent, a large available surface area is desirable, which is possible if it has been in dispersed condition (higher $\xi$ value). The $\xi$ and hydrodynamic radius ($R_h$) values of DTAB-rich and SDS-rich surfactants with and without MO and MB with an aqueous medium have been determined. The $\xi$, $R_h$ and PDI values (table 8) reflect the dispersion and stability of the solution. The higher $\xi$ value indicates the higher repulsive strength of molecules [82]. Our study found the higher positive $\xi$ value (27.02 mV) with MB + water system, whereas the lower $\xi$ value (0.53 mV) with 0.005 M DTAB+MB system; similarly, a higher negative $\xi$ values $-15.13$ with 0.012 M SDS + 0.005 M DTAB, while the lower $\xi$ value $-0.54$ with MO + water system.

The PDI is a dimensionless parameter. If the PDI values are less than 0.05, it means that the solution has a high monodispersity, while for greater than 0.7, it indicates that the solution has a polydispersive nature [83]. The interaction between MB and SDS indicates that MB is encountering a microenvironment in the SDS micelles. Probably, the SDS-MB form stable solution which could be induced by both coulombic and hydrophobic interactions [84], and the interaction of MB [85] with SDS could proceed by the mechanism opposite to that of micelle formation. In our study, MB and SDS concentrations being unchanged, the initially clear, homogenous solution becomes opalescent. Similar changes (appearance of opalescence) were also observed in ageing of initially homogeneous SDS-MB solutions, suggesting that the system is thermodynamically and kinetically stable. So, SDS-MB is strongly monodispersive with highly dispersive solutions. In the case of DTAB-rich samples as 0.005 M DTAB + MB, 0.005 M DTAB + MO are also highly monodispersive with strongly stable and highly dispersive solutions.

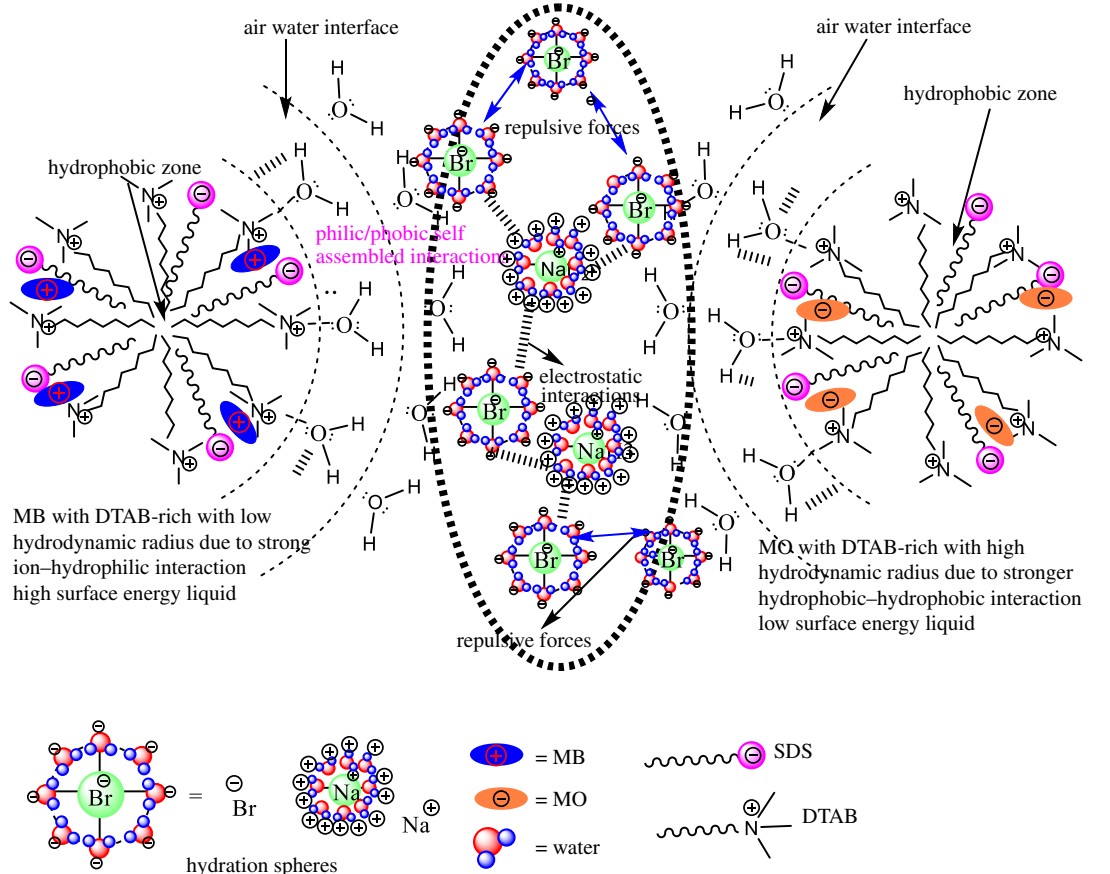

**Figure 13.** Molecular structural interactions of MO and MB with DTAB-rich mixed surfactants.

**Table 8.** Zeta potential ($\xi$), hydrodynamic radius ($R_h$) and PDI value of DTAB-rich and SDS-rich in the presence and absence of MB and MO at 298.15 K.

| systems | $R_h$ (nm) | PDI | $\xi$ (mV) |
|---|---|---|---|
| SDS-rich | | | |
| 0.01 M SDS | 3640 | 1.201 | 0.57 |
| 0.012 M SDS + 0.005 M DTAB + MB | 3720 | 1.170 | −11.10 |
| 0.01 M SDS + MB | 5230 | 0.1152 | 1.75 |
| MO + water | 5160 | 0.2680 | −0.54 |
| 0.01 M SDS + MO | 4650 | 1.784 | 0.56 |
| 0.012 M SDS + 0.005 M DTAB | 3270 | 1.708 | −15.13 |
| 0.012 MSDS + 0.005 M DTAB + MO | 3630 | 0.706 | −13.75 |
| DTAB-rich | | | |
| 0.005 M DTAB | 2355 | 1.589 | 0.55 |
| 0.00504 M DTAB + 0.01 M SDS + MB | 3530 | 1.0310 | 12.59 |
| 0.005 M DTAB + MB | 4660 | 0.392 | 0.53 |
| MB + water | 4780 | 1.846 | 27.02 |
| 0.005 M DTAB + MO | 5350 | 0.2394 | 0.55 |
| 0.00504 M DTAB + 0.01 M SDS | 5990 | 2.285 | 14.18 |
| 0.00504 M DTAB + 0.01 M SDS + MO | 6000 | 2.481 | −9.15 |

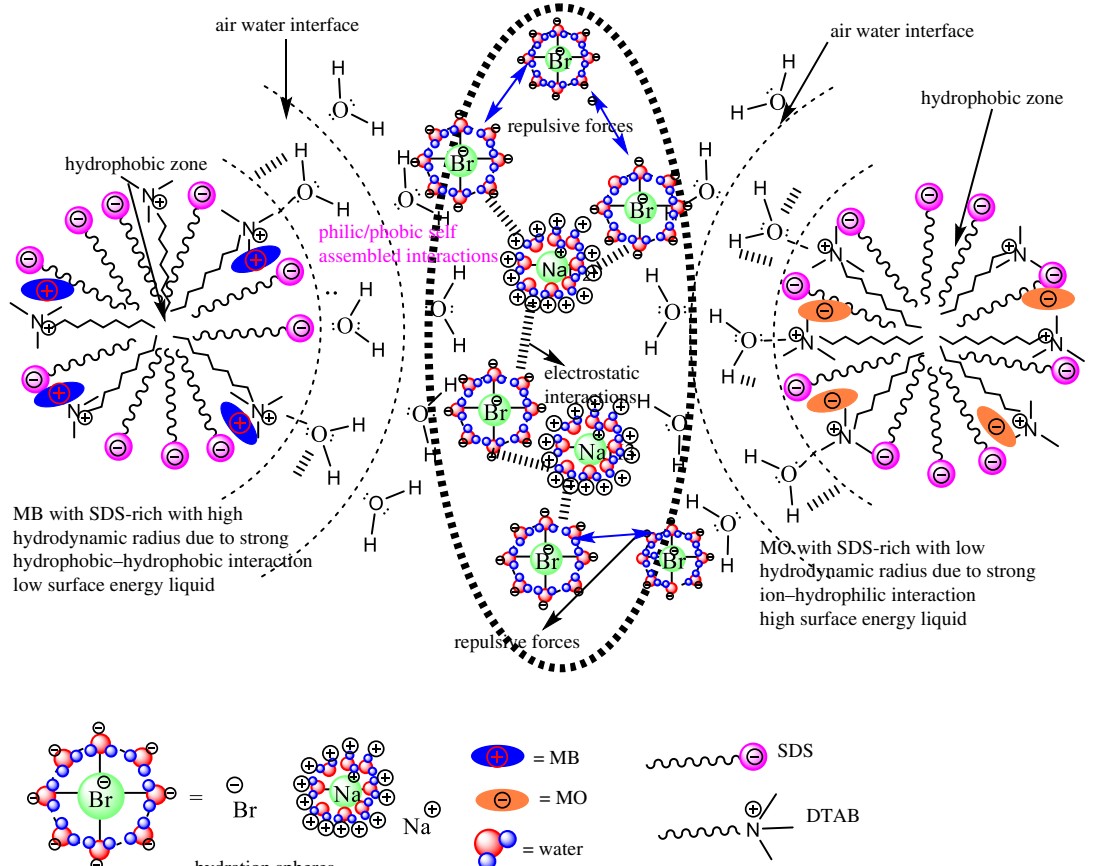

**Figure 14.** Molecular structural interactions of MO and MB with SDS-rich mixed surfactants.

Similarly, the interaction mechanism could be developed in the DTAB-MO system; DTAB and MO depict that the stronger electrostatic interactions for positive part of quaternary ammonia of DTAB and negative part of a sulfonate group of the MO with hydrophobic interacctions could be induced in the electrostatic and ion-phobic/philic interactions with DTAB-MO system. The maximum PDI value is 2.481 for 0.00504 M DTAB + 0.01 M SDS + MO and the minimum PDI value is 0.2394 for 0.005 M DTAB + MO.

The surfactants and dyes are developed in an individual hydration sphere in the solution. Due to the stronger electrostatic, ion-dipole and intermolecular interaction, this could be started as the formation of stable dye-surfactant solution due to such interaction; the dyes are arranged like H-type aggregation [86]. It depicted that the dye-surfactant is a monomer with the electrostatic interaction among positive part of the quaternary nitrogen of DTAB and negative part of sulphonate group, and the alkyl chain length (ACL) of DTAB strongly interacts with azo-group (chromophoric unit) of the rest of the molecules of MO dye [87]. Moreover, lots of factors such as mobility and dispersivity play an essential role in the dye–surfactant interaction. The H-type aggregation of MO with DTAB molecules in the liquid mixture and also different types of aggregates could be dependent on the ACL. The methylene groups $(-CH_2-)$ in the ACL could affect the interacting and packing parameters of dye-surfactant in the aqueous medium. The hydrophobic nature of ACL also has an important impact on the DTAB aggregation. Thus, the ion–hydrophobic, electrostatic and ion–dipole interactions (IDI) in between dye and surfactants micelles play a significant part in the penetration of dye into micelles and the solubilization and distribution of the dye molecules between aqueous and micellar phases. Table 8 shows that the SDS-rich system 0.012 M SDS + 0.005 M DTAB has (3270 nm) the lowest hydrodynamic radius due to the stronger electrostatic interaction and intermolecular forces, while the 0.005 M DTAB system shows a stronger interaction with solvent molecules. The DTAB-rich system 0.00504 M DTAB + 0.01 M SDS + MO has (6000 nm) the highest hydrodynamic radius. It depicts that due to the ion–hydrophobic interaction dominant over IHI.

Figure 13 shows that an addition of MB into DTAB-rich mixed surfactant shows weak interaction because DTAB and MB have same charges, due to the dominant ion-hydrophilic interaction (IHI) over the ion-hydrophobic interaction $(IH_bI)$ with decreasing $R_h$ value. Similarly, with MO into DTAB-rich, the $R_h$ value is increased with strong hydrophobic-hydrophobic interaction $(H_bH_bI)$.

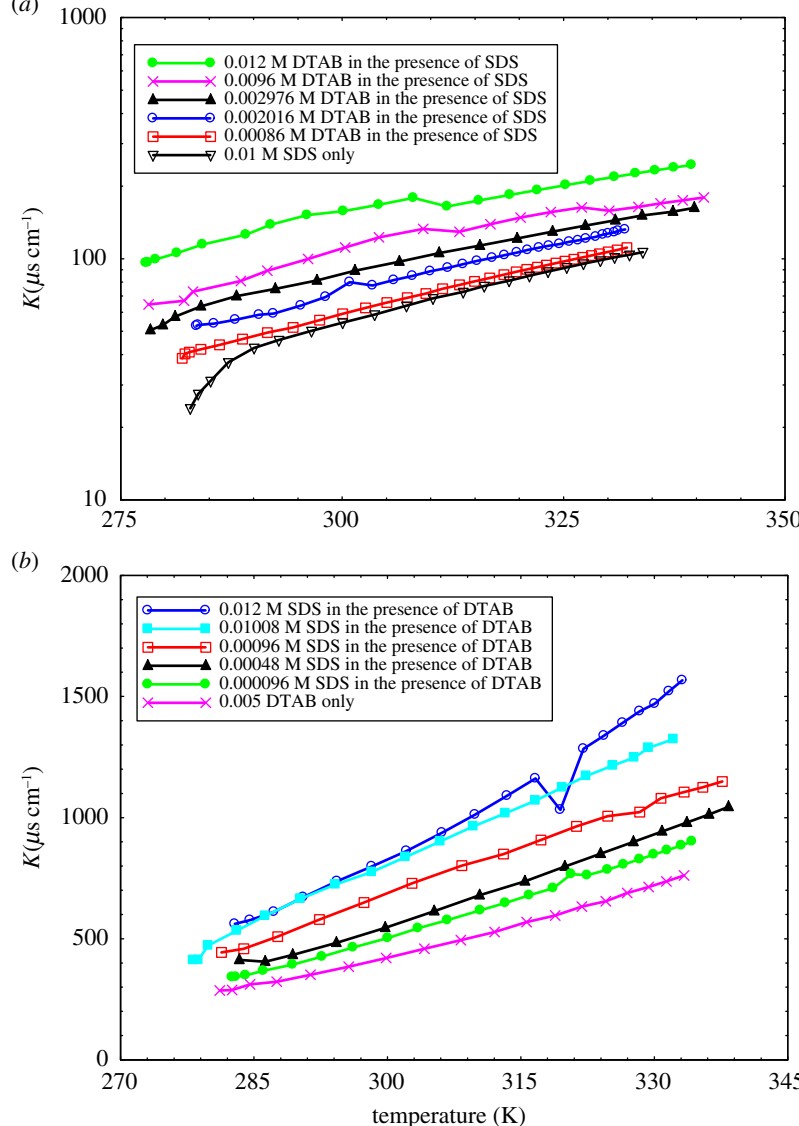

**Figure 15.** Conductivity ($\kappa$) versus T/K for Krafft temperature ($T_K$) of (a) DTAB-rich and (b) SDS-rich surfactants.

MB with SDS-rich mixed surfactant (figure 14) shows strong $H_bH_bI$ because SDS and MB have phobic/philic nature, it could be induced by weak electrostatic interaction and increased the $R_h$ value with high surface energy liquids, while with MO into SDS-rich (figure 14), the $R_h$ value is decreased because of strong electrostatic interaction and multiple intermolecular interactions. On addition of SDS and DTAB into aq-DTAB and aq-SDS solution, the increased $R_h$ size of mixed surfactant is apparently increased as the size of aggregates increased (table 8) and also, by a DLS experiment wherein the $R_h$ value of the DTAB-rich and SDS-rich were observed to increase in the presence of MO and MB dyes (table 8). This is in contrast to the stability of the $R_h$ in the absence and presence of dyes with a mixed surfactant system. In our study, we observed an inclusion of MO and MB into SDS-rich and DTAB-rich mixed surfactant; the $R_h$ is increased as well as aggregation increases with the higher stability of the solution. The systems containing ionic dyes and surfactants charged opposite to the dye electrostatic interactions develop. As a result of the attraction forces, ionic pairs dye-surfactant (MB and MO with DTAB-rich and SDS-rich) are formed in a stable solution [88]. Hydrophobic interactions, electrostatic interactions, hydrogen bonds, p-stacking and Van der Waals forces are typical examples of the intermolecular forces that dominate the interactions of dye molecules with surfactant aggregates [89].

## 3.5. Krafft point measurements

The Krafft point is generally the interaction between the solubility and CMC of the surfactant [18]. With increasing temperature, the solubility is increased. The solubility is adequate to the CMC at a particular

temperature, and hence the micelles can form [90]. The solubility and CMC curves intersect at the specific temperature called the Krafft temperature ($T_K$). Figure 15 of SDS-rich and DTAB-rich were measured by the use of $\kappa$ data in the aqueous medium [88].

The degree of counterion dissociation of ionic micelles is calculated from $T_K$ measurements. $T_K$ of DTAB and SDS are the roles of the counterions concentration in water. When the aqueous solution of ionic surfactant goes above the saturation concentration, a hydrated solid surfactant stage is divided [89]. The solubility is determined at temperatures below the Krafft point. The micelles begin to form at the Krafft point. The Krafft point can be determined as the temperature at which the solubility versus temperature curve intersects. Zhang *et al.* [91] have reported Krafft point and the solubility of SDS only on the function of temperature. By the use of the third-order polynomial equation, the relationship between solubility and temperature of surfactants has been determined. Gayet *et al.* [92] have studied the temperature effect on CMC value for SDS. An addition of MB and MO into SDS-rich and DTAB-rich mixed surfactant could induce electrostatic interactions; multiple intermolecular interactions (IMMI) form the stable thermodynamic solution. Although for cationic-anionic combinations a high probability of precipitation through change neutralization at comparable ratio is present, when one component is increased, the stably mixed micelles are usually formed [93]. However, in this study, we have chosen 3 : 1 critical ratio of surfactants; at this ratio surfactants do not form precipitation with and without dyes. Figure 15 illustrates that with increasing DTAB and SDS concentration, the $T_K$ of DTAB-rich and SDS-rich was first increased rapidly and then drastically decreases. It is due to the electrostatic repulsion forms the local concentration of DTAB-rich and SDS-rich increase in the presence of SDS/DTAB in an aqueous medium. Therefore, DTAB-rich and SDS-rich mixed surfactants have no significant effect on the solubility of SDS and DTAB in liquid water at temperatures between 313.15 and 323.15 K, as shown in figure 15. Also, there is no remarkable rise in the solubility before temperature move towards the minimum temperature of the normal Krafft point 313.15 and 323.15 K. This also indicates that the Krafft point for DTAB-rich and SDS-rich mixed surfactants show different interaction activities, this condition does not shift to a temperature below 313.15 and 323.15 K.

# 4. Conclusion

The conductance and surface tension of SDS-rich and DTAB-rich mixtures in an aqueous medium at $T =$ 293.15, 298.15 and 303.15 K were used as a function of surfactant concentration. On increasing the concentration of surfactant and temperature, the conductance values are increased and CMC and $\alpha$ also increased. The CMC obtained by conductance and surface tension are close to each other. Also, $\Delta G_m^o$ and $\Delta H_m^o$ are found more negative with increment in temperature, while $\Delta S_m^o$ is found decreased with increase in temperature. The Krafft point is determined as the temperature where the solubility against temperature plot intersects with the specific conductance versus temperature graph. The surface tension is decreased at first with increment in the concentration of surfactant mixtures in water. DTAB-rich systems show the antagonistic interaction, whereas SDS-rich systems explore synergistic interaction. Hence the synergism in mixed micelle formation is present when mixed surfactants have a lower CMC than the individual surfactants. When the CMC is increased, then the value of $\Gamma_{max}$ decreases and $A_{min}$ followed the opposite trend. Similarly, when the CMC is increased, $\pi_{cmc}$ increases while $P$ and $G_{ads}^o$ decreases. The $\xi$ and PDI values are reflected in the dispersion and stability of the solution. An addition of dyes into SDS and DTAB-rich system increases the monodispersibility, which is confirmed by PDI measurement. The micellar size distribution of DTAB and SDS-rich surfactant analysed by DLS has confirmed their effective micellization for the stability of the solution with and without dyes. The binding and distribution constant of MO and MB between the aqueous phase and DTAB-rich micellar phase have been calculated efficiently.

Data accessibility. The graphical data of specific conductivities and UV–Vis that supporting this paper have been uploaded as electronic supplementary material.

Authors' contributions. K.M.S. and S.A.K. conducted the experiments and data analysis, M.S. guided the writing process and A.B. wrote the paper. All the authors gave their final approval for publication.

Competing interests. The authors declare no competing interest.

Funding. This study was supported by The World Academy of Sciences, Italy under the Award of 2016 TWAS Fellowship for Research and Advanced Training FR number: 3240288440 to Dr Ajaya Bhattarai.

Acknowledgements. We thank the editor and anonymous reviewers for their valuable comments and suggestions.

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
