## [Reviewer comments · Royal Society Open Science]

Review History

RSOS-181979.R0 (Original submission)

Review form: Reviewer 1

Is the manuscript scientifically sound in its present form?

No

Are the interpretations and conclusions justified by the results?

No

Is the language acceptable?

Yes

Is it clear how to access all supporting data?

Not Applicable

Do you have any ethical concerns with this paper?

No

Have you any concerns about statistical analyses in this paper?

Yes

Recommendation?

Reject

Comments to the Author(s)

This is a poor paper of limited scope and impact, and is not suitable for publication in the Royal Society Open Science.

The paper presents and discusses data on SDS / DTAB mixtures at a superficial level. Surfactant mixing, even in the presence of other additives, has been extensively studied and reported; and this includes anionic / cationic mixtures. This paper adds nothing of significance to that extensive literature.

Furthermore the authors seem unaware of the recent developments in advanced experimental techniques to study surfactant mixing at interfaces and in micelles, and use only a rather dated and limited approach. They also seem unaware of the developments in the application of the pseudo phase approximation to surfactant mixing, and make no attempt to quantify the extent of the synergy in the mixing.

The derived data in table 4 contains quite unrealistic errors, and indicate lack of critical evaluation of the data.

It is not at all clear what the DLS measurements indicate and how they are interpreted in terms of surfactant self-assembly.

To even publish this work in a more specialist and appropriate journal would require an enormous amount of re-evaluation and revision.

Review form: Reviewer 2

Is the manuscript scientifically sound in its present form?

Yes

Are the interpretations and conclusions justified by the results?

Yes

Is the language acceptable?

Yes

Is it clear how to access all supporting data?

Yes

Do you have any ethical concerns with this paper?

No

Have you any concerns about statistical analyses in this paper?

No

Recommendation?

Accept with minor revision (please list in comments)

Comments to the Author(s)

Appropriately reducing the number of references

Review form: Reviewer 3

Is the manuscript scientifically sound in its present form?

Yes

Are the interpretations and conclusions justified by the results?

No

Is the language acceptable?

No

Is it clear how to access all supporting data?

Yes

Do you have any ethical concerns with this paper?

No

Have you any concerns about statistical analyses in this paper?

No

Recommendation?

Accept with minor revision (please list in comments)

Comments to the Author(s)

Adjudicative reviewer's comments:

This manuscript (RSOS 181979) authored by K. M. Sachin, et al reports their study on a cationic surfactant system consisting of DTAB and SDS. They have studied several characteristics of this mixed surfactant system such as CMC, Krafft point, dispersibility and stability of the SDS-rich and DTAB-rich solutions in the presence and absence of dyes, methyl orange (MO) and methylene blue (MB). In this sense, it's a paper of broad scope and can make a notable contribution in the science of surfactants. They have fixed their research question based on an exhaustive literature survey and the subject seems to be a relevant one. The change in CMC value as observed in their study supports the assertion that this mixed surfactant system can have practical applications. They have employed appropriate techniques, e.g., surface tension measurement, conductivity measurement, spectrophotometry, dynamic light scattering, etc. for their study. Therefore, scientifically, this paper is sound. But it possesses several grammatical/writing style flaws which the authors must correct. I also suggest a careful revision of the technical aspects stated in my comments, given below, and expect appropriate answers to the questions raised there. I recommend publishing this article in the journal Royal Society Open Science once the authors make an appropriate revision.

Comments related to the subject matter

1. Page 3, Line 28, 29- What does "micellization of Gibb's free energy" mean? Is it "Gibb's free energy of micellization" instead? Please check.
2. Page 3, Line 36 to 46- It is understandable that there is a lack of literature related to the effects of dyes on these surfactants. Apart from that, what are the other factors that necessitate such a study, i.e., why the effects of dyes on these surfactant systems need to be known; is it for the need of industries or some other practical reasons? If there is a practical reason for enriching literature in this subject, please write.
3. Page 4- Table 1 in the "Materials" section seems unnecessary. Please remove it unless just mentioning purities of the chemicals is not enough.
4. Page 8, Line 45-57- CMC of both SDS-rich and DTAB-rich formulations seem to be less than

that of the individual surfactants. It seems to be an important character. But a discussion regarding the significance of this change is lacking. The conclusion of the paper also does not include this aspect.

5. Page 14, Line 24- Synergistic effect, which is a desirable character in surfactant mixtures, from the practical viewpoint, is mentioned in this line. But its broader discussion is lacking in the manuscript and the conclusion also does not put an emphasis on it.

6. Page 11, Line 17-22- It is written that Amin increases from 62.10 Å² to 49.70 Å². Please check it.

7. Page 12, Line 14 and Line 49- Spelling of the cited author's name is wrong.

8. Page 21, Line 10-14- Is the finding mentioned in these lines your own finding or that of reference 91? Please write more clearly.

9. Page 21, Line 35-38- These lines, combined with Line 42-45 on page 24 means that the interaction between MO and DTAB-rich solution is weak, but still, it is stronger than the interaction between MB and DTAB-rich. Please write it more clearly so that these two statements do not sound contradictory to each other.

10. DTAB forms complex with MO which is not a desirable character for the functioning of surfactants. You seem to have tried to justify this on page 29 and 30, but the statement is unclear. Please discuss more clearly, why, despite the formation of a complex, this formulation is important.

11. Study of the effects of MO and MB is not properly justified. What notable properties were observed in these studies? Was there a synergistic or antagonistic interaction, or anything else that is interesting?

12. Elaborate how the ζ potential and hydrodynamic diameter provide information about the stability and dispersibility of these formulations. You may enrich your discussion with the help of some recent publications, e.g., those on phospholipids containing a charged additive.

13. Please make your conclusion more informative by incorporating several aspects of the study that are discussed in the document. Some examples are as follows-

i. How CMC of the system changes compared to the individual surfactant solutions.

ii. How do the ζ potential and PDI values reflect the stability of dispersion?

iii. How does the stability of the solutions with and without dyes compare with each other?

14. The term "matches with the literature" is used too often while citing literatures. Please find appropriate substitutes, e.g., "agrees with...", "same as...", "similar to...", "corresponds to...", "close to...", etc.

15. If possible, limit the number of author names to fewer in citations and use only numbers for that purpose as long as possible.

Comments related to language

16. Please consider revising the following sentences for grammatical/ writing-style flaws:

Line 17-18, page 2: "The synergism happens for the formation of mixed micelle as the CMC of mixed systems is less than to the pure system." is unclear or wrong.

Line 9-12, page 3:

The sentence "The researchers have been reported on the interaction study of the mixed surfactants in the presence of salts in different medium by using several methods." is unclear and wrong.

Line 16-17, page 3- The sentence "But they were not reported the effect of temperature on CMC as well as all the thermodynamics and surface properties." is wrong, consider revising it.

Line 37, page 3- The sentence "Earlier studies have been reported the effect of dyes on the single surfactant" is wrong, consider revising it.

Line 37-40, page 3- The sentence

“Samiey and Ashoori [54] have been determined only the kinetic and thermodynamic studies in the presence of crystal violet with DTAB and SDS separately.” is wrong, correct it; also correct each sentence following it in the same paragraph.

Page 12, Line 53-58- The sentence “Nevertheless, with increasing the temperature, the P values decreases for both DTAB-rich and SDS-rich system because of increasing the kinetic energy of the molecules which could be induced oscillation (vibrational, rotational and translational) and decrease the binding forces [67].” is wrong. Please avoid the use of passive voice in such an inappropriate way at other places too.

Page 13, Line 17-19- The sentence “But the value of ΔG_{ads} for SDS-rich in water is found to be decrease as $-45.29 \text{ kJmol}^{-1}$ (table 3) at 298.15 K.” is wrong.

Page 13, Line 26- Here and at all other places, please keep a space on both sides of the equality sign (=).

Page 15, Line 53, 54- The sentence “The more negative of ΔG_{om} means the more spontaneous and micellization will feasible” is wrong.

17. Also, carefully revise the document for other language-related errors by yourself.

Comments as an adjudicative reviewer.

1. I agree with reviewer-2's comment.

2. Regarding reviewer-1's comments, I have following opinions.

- i. Regarding the scope and impact of the paper, since it covers a study on a broad range of parameters, I think its scope is not so limited if a proper revision is done.
- ii. Regarding the superficial level of discussion on the data, if they incorporate the recommendations mentioned above, I hope this will be improved.
- iii. Regarding synergy, DLS measurements, etc., I have mentioned in my comments and if they are able to revise appropriately, these problems can be solved.
- iv. I suggest the authors to consider the points raised by reviewer-1, too, while revising their manuscript.
- v. Regarding appropriacy of the manuscript for the journal, I think it is appropriate to publish after revision.

Decision letter (RSOS-181979.R0)

06-Feb-2019

Dear Dr Bhattarai:

Title: Self-assembly of SDS and DTAB mixed surfactants with dyes in aqueous mixtures
Manuscript ID: RSOS-181979

The editor assigned to your manuscript has now received comments from reviewers. We would like you to revise your paper in accordance with the referee and Subject Editor suggestions which

can be found below (not including confidential reports to the Editor). Please note this decision does not guarantee eventual acceptance.

Please submit your revised paper before 01-Mar-2019. Please note that the revision deadline will expire at 00.00am on this date. If we do not hear from you within this time then it will be assumed that the paper has been withdrawn. In exceptional circumstances, extensions may be possible if agreed with the Editorial Office in advance. We do not allow multiple rounds of revision so we urge you to make every effort to fully address all of the comments at this stage. If deemed necessary by the Editors, your manuscript will be sent back to one or more of the original reviewers for assessment. If the original reviewers are not available we may invite new reviewers.

Please also include the following statements alongside the other end statements. As we cannot publish your manuscript without these end statements included, if you feel that a given heading is not relevant to your paper, please nevertheless include the heading and explicitly state that it is not relevant to your work.

- Ethics statement

Please clarify whether you received ethical approval from a local ethics committee to carry out your study. If so please include details of this, including the name of the committee that gave consent in a Research Ethics section after your main text. Please also clarify whether you received informed consent for the participants to participate in the study and state this in your Research Ethics section.

OR

Please clarify whether you obtained the necessary licences and approvals from your institutional animal ethics committee before conducting your research. Please provide details of these licences and approvals in an Animal Ethics section after your main text.

OR

Please clarify whether you obtained the appropriate permissions and licences to conduct the fieldwork detailed in your study. Please provide details of these in your methods section.

Royal Society of Chemistry
Thomas Graham House
Science Park, Milton Road

Cambridge, CB4 0WF
Royal Society Open Science - Chemistry Editorial Office

RSC Associate Editor:
Comments to the Author:
(There are no comments.)

RSC Subject Editor:
Comments to the Author:
(There are no comments.)

Reviewers' Comments to Author:
Reviewer: 1

Comments to the Author(s)

This is a poor paper of limited scope and impact, and is not suitable for publication in the Royal Society Open Science.

The paper presents and discusses data on SDS / DTAB mixtures at a superficial level. Surfactant mixing, even in the presence of other additives, has been extensively studied and reported; and this includes anionic / cationic mixtures. This paper adds nothing of significance to that extensive literature.

Furthermore the authors seem unaware of the recent developments in advanced experimental techniques to study surfactant mixing at interfaces and in micelles, and use only a rather dated and limited approach. They also seem unaware of the developments in the application of the pseudo phase approximation to surfactant mixing, and make no attempt to quantify the extent of the synergy in the mixing.

The derived data in table 4 contains quite unrealistic errors, and indicate lack of critical evaluation of the data.

It is not at all clear what the DLS measurements indicate and how they are interpreted in terms of surfactant self-assembly.

To even publish this work in a more specialist and appropriate journal would require an enormous amount of re-evaluation and revision.

Reviewer: 2

Comments to the Author(s)

Appropriately reducing the number of references

Reviewer: 3

Comments to the Author(s)

Adjudicative reviewer's comments:

This manuscript (RSOS 181979) authored by K. M. Sachin, et al reports their study on a cationic surfactant system consisting of DTAB and SDS. They have studied several characteristics of this mixed surfactant system such as CMC, Krafft point, dispersibility and stability of the SDS-rich

and DTAB-rich solutions in the presence and absence of dyes, methyl orange (MO) and methylene blue (MB). In this sense, it's a paper of broad scope and can make a notable contribution in the science of surfactants. They have fixed their research question based on an exhaustive literature survey and the subject seems to be a relevant one. The change in CMC value as observed in their study supports the assertion that this mixed surfactant system can have practical applications. They have employed appropriate techniques, e.g., surface tension measurement, conductivity measurement, spectrophotometry, dynamic light scattering, etc. for their study. Therefore, scientifically, this paper is sound. But it possesses several grammatical/writing style flaws which the authors must correct. I also suggest a careful revision of the technical aspects stated in my comments, given below, and expect appropriate answers to the questions raised there. I recommend publishing this article in the journal Royal Society Open Science once the authors make an appropriate revision.

Comments related to the subject matter

1. Page 3, Line 28, 29- What does "micellization of Gibb's free energy" mean? Is it "Gibb's free energy of micellization" instead? Please check.
2. Page 3, Line 36 to 46- It is understandable that there is a lack of literature related to the effects of dyes on these surfactants. Apart from that, what are the other factors that necessitate such a study, i.e., why the effects of dyes on these surfactant systems need to be known; is it for the need of industries or some other practical reasons? If there is a practical reason for enriching literature in this subject, please write.
3. Page 4- Table 1 in the "Materials" section seems unnecessary. Please remove it unless just mentioning purities of the chemicals is not enough.
4. Page 8, Line 45-57- CMC of both SDS-rich and DTAB-rich formulations seem to be less than that of the individual surfactants. It seems to be an important character. But a discussion regarding the significance of this change is lacking. The conclusion of the paper also does not include this aspect.
5. Page 14, Line 24- Synergistic effect, which is a desirable character in surfactant mixtures, from the practical viewpoint, is mentioned in this line. But its broader discussion is lacking in the manuscript and the conclusion also does not put an emphasis on it.
6. Page 11, Line 17-22- It is written that Amin increases from 62.10 Å² to 49.70 Å². Please check it.
7. Page 12, Line 14 and Line 49- Spelling of the cited author's name is wrong.
8. Page 21, Line 10-14- Is the finding mentioned in these lines your own finding or that of reference 91? Please write more clearly.
9. Page 21, Line 35-38- These lines, combined with Line 42-45 on page 24 means that the interaction between MO and DTAB-rich solution is weak, but still, it is stronger than the interaction between MB and DTAB-rich. Please write it more clearly so that these two statements do not sound contradictory to each other.
10. DTAB forms complex with MO which is not a desirable character for the functioning of surfactants. You seem to have tried to justify this on page 29 and 30, but the statement is unclear. Please discuss more clearly, why, despite the formation of a complex, this formulation is important.
11. Study of the effects of MO and MB is not properly justified. What notable properties were observed in these studies? Was there a synergistic or antagonistic interaction, or anything else that is interesting?
12. Elaborate how the ζ potential and hydrodynamic diameter provide information about the stability and dispersibility of these formulations. You may enrich your discussion with the help of some recent publications, e.g., those on phospholipids containing a charged additive.
13. Please make your conclusion more informative by incorporating several aspects of the study that are discussed in the document. Some examples are as follows-
 - i. How CMC of the system changes compared to the individual surfactant solutions.
 - ii. How do the ζ potential and PDI values reflect the stability of dispersion?
 - iii. How does the stability of the solutions with and without dyes compare with each other?

14. The term “matches with the literature” is used too often while citing literatures. Please find appropriate substitutes, e.g., “agrees with...”, “same as...”, “similar to...”, “corresponds to...”, “close to...”, etc.

15. If possible, limit the number of author names to fewer in citations and use only numbers for that purpose as long as possible.

Comments related to language

16. Please consider revising the following sentences for grammatical/ writing-style flaws:

Line 17-18, page 2: “The synergism happens for the formation of mixed micelle as the CMC of mixed systems is less than to the pure system.” is unclear or wrong.

Line 9-12, page 3:

The sentence “The researchers have been reported on the interaction study of the mixed surfactants in the presence of salts in different medium by using several methods.” is unclear and wrong.

Line 16-17, page 3- The sentence “But they were not reported the effect of temperature on CMC as well as all the thermodynamics and surface properties.” is wrong, consider revising it.

Line 37, page 3- The sentence “Earlier studies have been reported the effect of dyes on the single surfactant” is wrong, consider revising it.

Line 37-40, page 3- The sentence

“Samiey and Ashoori [54] have been determined only the kinetic and thermodynamic studies in the presence of crystal violet with DTAB and SDS separately.” is wrong, correct it; also correct each sentence following it in the same paragraph.

Page 12, Line 53-58- The sentence “Nevertheless, with increasing the temperature, the P values decreases for both DTAB-rich and SDS-rich system because of increasing the kinetic energy of the molecules which could be induced oscillation (vibrational, rotational and translational) and decrease the binding forces [67].” is wrong. Please avoid the use of passive voice in such an inappropriate way at other places too.

Page 13, Line 17-19- The sentence “But the value of ΔG_{ads} for SDS-rich in water is found to be decrease as $-45.29 \text{ kJmol}^{-1}$ (table 3) at 298.15 K.” is wrong.

Page 13, Line 26- Here and at all other places, please keep a space on both sides of the equality sign (=).

Page 15, Line 53, 54- The sentence “The more negative of ΔG_{om} means the more spontaneous and micellization will feasible” is wrong.

17. Also, carefully revise the document for other language-related errors by yourself.

Comments as an adjudicative reviewer.

1. I agree with reviewer-2's comment.
2. Regarding reviewer-1's comments, I have following opinions.

- i. Regarding the scope and impact of the paper, since it covers a study on a broad range of parameters, I think its scope is not so limited if a proper revision is done.
- ii. Regarding the superficial level of discussion on the data, if they incorporate the recommendations mentioned above, I hope this will be improved.
- iii. Regarding synergy, DLS measurements, etc., I have mentioned in my comments and if they are able to revise appropriately, these problems can be solved.
- iv. I suggest the authors to consider the points raised by reviewer-1, too, while revising their manuscript.
- v. Regarding appropriacy of the manuscript for the journal, I think it is appropriate to publish after revision.

Author's Response to Decision Letter for (RSOS-181979.R0)

See Appendix A.

Decision letter (RSOS-181979.R1)

01-Mar-2019

Dear Dr Bhattarai:

Title: Self-assembly of SDS and DTAB mixed surfactants with dyes in aqueous mixtures
Manuscript ID: RSOS-181979.R1

It is a pleasure to accept your manuscript in its current form for publication in Royal Society Open Science. The chemistry content of Royal Society Open Science is published in collaboration with the Royal Society of Chemistry.

RSC Associate Editor
Comments to the Author:
(There are no comments.)

Reviewer(s)' Comments to Author:

Appendix A

Response to reviewer 1

We have been highly benefited by the valuable expressions of the reviews about the quality of our paper.

The paper presents and discusses data on SDS / DTAB mixtures at a superficial level. Surfactant mixing, even in the presence of other additives, has been extensively studied and reported; and this includes anionic / cationic mixtures. This paper adds nothing of significance to that extensive literature.

Ans: We are very happy with the observation of the respectful reviewer who has deep knowledge in our fields. So, we have done corrections in the revised manuscript.

Furthermore the authors seem unaware of the recent developments in advanced experimental techniques to study surfactant mixing at interfaces and in micelles, and use only a rather dated and limited approach. They also seem unaware of the developments in the application of the pseudo phase approximation to surfactant mixing, and make no attempt to quantify the extent of the synergy in the mixing.

Ans We agreed with the suggestions given by the respectful reviewer and applied following explanation in our revised manuscript which are given below:

On the explanation for synergetic effect, we tried our mixed CMC data by coupling the pseudo phase separation model with a regular solution approximation which was proposed by Rubingh [Holland, P. M.; Rubingh, D. N. J. Phys. Chem. 1983, 87, 1984– 1990] and Rosen et al [Rosen, M. J.; Hua, X. Y. J. Colloid Interface Sci. 1982, 86, 164– 172] and was not able to give completely satisfactory results for mixed CMC. So, we did not try on the model of Motomura et al [Motomura, K.; Yamanaka, M.; Aratono, M. Colloid Polym. Sci. 1984, 262, 948–955] though the model is in the view of thermodynamics and also takes into account dissociation of ionic surfactants but the model does not deal with the surfactant interaction parameter.

Eventually, we got success to apply the theoretical model that is the pseudo phase separation model which is coupled with the dissociated Margules Model [Eads, C. D.; Robosky, L. C. *Langmuir* 1999, 15, 2661–2668, Hu, J.; Zhou, L.; Feng, J.; Liu, H.; Hu, Y. J. *Colloid Interface Sci.* 2007, 315, 761–767, Hao, L.-S *et al.*, *The Journal of Physical Chemistry B*, 2012, 116(17), 5213–5225] which gives satisfactory description of the mixed CMC as well as surfactant interaction parameter.

The derived data in table 4 contains quite unrealistic errors, and indicate lack of critical evaluation of the data.

Ans: We have corrected and incorporated in the revised manuscript.

It is not at all clear what the DLS measurements indicate and how they are interpreted in terms of surfactant self-assembly.

Ans: We really appreciate the valuable note of the reviewer and we have corrected and adequately incorporated needful science in the manuscript. Such guidelines from reviewer really make the new landmarks for fitting our paper in most scientific ensembles with high impact and self-explanatory meaning and understanding.

To even publish this work in a more specialist and appropriate journal would require an enormous amount of re-evaluation and revision.

Ans: We have done enormous correction and revision in each and every aspects of our revised manuscript.

Response to reviewer 2

1. Appropriately reducing the number of references

Ans: We have reduced the number of references in the revised manuscript.

Response to reviewer 3

Comments related to the subject matter

1. Page 3, Line 28, 29- What does “micellization of Gibb’s free energy” mean? Is it “Gibb’s free energy of micellization” instead? Please check.

Ans: It will be mistake if we use micellization of Gibb's free energy instead of Gibb's free energy of micellization . So, the needful corrections are made and are incorporated in the revised manuscript.

2. Page 3, Line 36 to 46- It is understandable that there is a lack of literature related to the effects of dyes on these surfactants. Apart from that, what are the other factors that necessitate such a study, i.e., why the effects of dyes on these surfactant systems need to be known; is it for the need of industries or some other practical reasons? If there is a practical reason for enriching literature in this subject, please write.

Ans: It has been a very effective suggestion and is now incorporated in the revised manuscript.

3. Page 4- Table 1 in the "Materials" section seems unnecessary. Please remove it unless just mentioning purities of the chemicals is not enough.

Ans: Corrections have incorporated in the revised manuscript.

4. Page 8, Line 45-57- CMC of both SDS-rich and DTAB-rich formulations seem to be less than that of the individual surfactants. It seems to be an important character. But a discussion regarding the significance of this change is lacking. The conclusion of the paper also does not include this aspect.

Ans: We have applied different models in the revised manuscript to explain the synergistic effect for mixed CMC of our systems and also included the findings in the conclusion section.

5. Page 14, Line 24- Synergistic effect, which is a desirable character in surfactant mixtures, from the practical viewpoint, is mentioned in this line. But its broader discussion is lacking in the manuscript and the conclusion also does not put an emphasis on it.

Ans: We really appreciate the valuable note of the referee and we have corrected and incorporated in the revised manuscript.

6. Page 11, Line 17-22- It is written that Amin increases from 62.10 Å² to 49.70 Å². Please check it.

Ans: These are very effective observations which are efficiently incorporated in the revised manuscript.

7. Page 12, Line 14 and Line 49- Spelling of the cited author's name is wrong.

Ans: Corrections have incorporated in the revised manuscript.

8. Page 21, Line 10-14- Is the finding mentioned in these lines your own finding or that of reference 91? Please write more clearly.

Ans: The finding mentioned in these lines (10-14 of Page 21) is not our findings but that was of reference 91. So, we have corrected our mistakes.

9. Page 21, Line 35-38- These lines, combined with Line 42-45 on page 24 means that the interaction between MO and DTAB-rich solution is weak, but still, it is stronger than the interaction between MB and DTAB-rich. Please write it more clearly so that these two statements do not sound contradictory to each other.

Ans: This observation of the reviewer has been an additional pressure on our minds to further relook and reveals the manifold hidden sciences out of such observations. The details new interpretations of the study are added in the text.

10. DTAB forms complex with MO which is not a desirable character for the functioning of surfactants. You seem to have tried to justify this on page 29 and 30, but the statement is unclear. Please discuss more clearly, why, despite the formation of a complex, this formulation is important.

Ans: We really appreciate the valuable note of the referee and we have corrected and adequately incorporated needful science in the manuscript.

11. Study of the effects of MO and MB is not properly justified. What notable properties were observed in these studies? Was there a synergistic or antagonistic interaction, or anything else that is interesting?

Ans: The working scientific modules and structural abilities derived from their structural abilities are incorporated in the revised text.

12. Elaborate how the ζ potential and hydrodynamic diameter provide information about the stability and dispersibility of these formulations. You may enrich your discussion with the help of some recent publications, e.g., those on phospholipids containing a charged additive.

Ans: Table 4 shows that the -1.708, -15.13 PDI and zeta potential value of SDS-rich mixed surfactant system. An addition of MB into SDS-rich system,

the PDI is decreased with stronger monodispersity and enhances the stability of the solution. It indicates that MB and DTAB both contain positive charged due to the cationic - cationic interaction dominant over the anionic-cationic interaction. Similar, addition of MO into SDS-rich system, the PDI value is more decreased with stronger monodispersity as well as potential value noticed more negative. It infers that the MO and SDS both are negative in nature, so in this situation DTAB showing stronger interacting ability than SDS with increased the PDI value with enhance higher monodispersity. However, due to the higher population of anions observed higher negative zeta potential. In case of DTAB-rich shows 2.285 PDI value with low monodispersity and 14.18 zeta potential. It depicted the lower population of DTAB ions and higher SDS population which could induce weak electrostatic interaction. An addition of MB into DTAB-rich the decrease of the PDI value because of MB and DTAB both are cationic in nature with shows higher monodispersity. Therefore, with MO the PDI is increased because of MO is anionic nature with weak electrostatic interaction as well as low monodispersity of the solution. It indicates that the due to the presence of higher anionic molecules, the surface charge obtained.

13. Please make your conclusion more informative by incorporating several aspects of the study that are discussed in the document. Some examples are as follows-

i. How CMC of the system changes compared to the individual surfactant solutions.

Ans: Corrections have been incorporated in the revised manuscript.

ii. How do the ζ potential and PDI values reflect the stability of dispersion?

Ans: Corrections have been incorporated in the revised manuscript.

iii. How does the stability of the solutions with and without dyes compare with each other?

Ans: These have been a valuable interface and are now adequately attempted.

14. The term “matches with the literature” is used too often while citing literatures. Please find appropriate substitutes, e.g., “agrees with...”, “same as...”, “similar to...”, “corresponds to...”, “close to...”, etc.

Ans: The reviewer has suggested very informative databases which are now incorporated in the revised manuscript.

15. If possible, limit the number of author names to fewer in citations and use only numbers for that purpose as long as possible.

Ans: Corrections have been incorporated in the revised manuscript.

Comments related to language

16. Please consider revising the following sentences for grammatical/writing-style flaws:

Line 17-18, page 2: “The synergism happens for the formation of mixed micelle as the CMC of mixed systems is less than to the pure system.” is unclear or wrong.

Ans: Corrections have been incorporated in the revised manuscript.

Line 9-12, page 3:

The sentence “The researchers have been reported on the interaction study of the mixed surfactants in the presence of salts in different medium by using several methods.” is unclear and wrong.

Ans: Corrections have been included in the revised manuscript.

Line 16-17, page 3- The sentence “But they were not reported the effect of temperature on CMC as well as all the thermodynamics and surface properties.” is wrong, consider revising it.

Ans: Corrections have been included in the revised manuscript.

Line 37, page 3- The sentence “Earlier studies have been reported the effect of dyes on the single surfactant” is wrong, consider revising it.

Ans: Corrections have been included in the revised manuscript.

Line 37-40, page 3- The sentence

“Samiey and Ashoori [54] have been determined only the kinetic and thermodynamic studies in the presence of crystal violet with DTAB and SDS separately.” is wrong, correct it; also correct each sentence

following it in the same paragraph.

Ans: Corrections have included in the revised manuscript.

Page 12, Line 53-58- The sentence “Nevertheless, with increasing the temperature, the P values decreases for both DTAB-rich and SDS-rich system because of increasing the kinetic energy of the molecules which could be induced oscillation (vibrational, rotational and translational) and decrease the binding forces [67].” is wrong. Please avoid the use of passive voice in such an inappropriate way at other places too.

Ans: Corrections have included in the revised manuscript.

Page 13, Line 17-19- The sentence “But the value of ΔG_{ads} for SDS-rich in

water is found to be decrease as $-45.29 \text{ kJmol}^{-1}$ (table 3) at 298.15 K.” is wrong.

Ans: Corrections have included in the revised manuscript.

Page 13, Line 26- Here and at all other places, please keep a space on both sides of the equality sign (=).

Ans: Corrections have included in the revised manuscript.

Page 15, Line 53, 54- The sentence “The more negative of ΔG_{om} means the more spontaneous and micellization will feasible” is wrong.

Ans: Corrections have incorporated in the revised manuscript.

17. Also, carefully revise the document for other language-related errors by yourself.

Ans: Corrections have included in the revised manuscript.